



# The importance of alkyl nitrates and sea ice emissions to atmospheric
# NOx sources and cycling in the summertime Southern Ocean marine
# boundary layer.
Jessica M. Burger[1], Julie Granger[2], Emily Joyce[3], Meredith G. Hastings[3], Kurt A.M. Spence[1], Katye E.
Altieri[1]
[1]Department of Oceanography, University of Cape Town, Rondebosch, 7701, South Africa
[2]Department of Marine Sciences, University of Connecticut, Groton, 06340, USA
[3]Department of Earth, Environmental and Planetary Sciences and Institute at Brown for Environment and Society, Brown
University, Providence, RI, 02906, USA.
*Correspondence to*: Jessica M. Burger (brgjes006@uct.ac.za)
**Abstract.** Atmospheric nitrate originates from the oxidation of nitrogen oxides ($NO_x=NO+NO_2$) and impacts both tropospheric
chemistry and climate. $NO_x$ sources, cycling, and $NO_x$ to nitrate formation pathways are poorly constrained in remote marine
regions, especially the Southern Ocean where pristine conditions serve as a useful proxy for the preindustrial atmosphere.
Here, we measured the isotopic composition ($\delta^{15}N$ and $\delta^{18}O$) of atmospheric nitrate in coarse-mode (> 1μm) aerosols collected
in the summertime marine boundary layer of the Atlantic Southern Ocean from 34.5°S to 70°S, and across the northern edge
of the Weddell Sea. The $\delta^{15}N\text{-}NO_3^-$ decreased with latitude from -2.7‰ to -43.1‰. The decline in $\delta^{15}N$ with latitude is
attributed to changes in the dominant $NO_x$ sources: lightning at the low latitudes, oceanic alkyl nitrates at the mid latitudes,
and photolysis of nitrate in snow at the high latitudes. There is no evidence of any influence from anthropogenic $NO_x$ sources
or equilibrium isotopic fractionation. Using air mass back trajectories and an isotope mixing model, we calculate that oceanic
alkyl nitrate emissions have a $\delta^{15}N$ signature of -22.0‰ ± 7.5‰. Given that measurements of alkyl nitrate contributions to
remote nitrogen budgets are scarce, this may be a useful tracer for detecting their contribution in other oceanic regions. The
$\delta^{18}O\text{-}NO_3^-$ was always less than 70‰, indicating that daytime processes involving OH are the dominant $NO_x$ oxidation pathway
during summer. Unusually low $\delta^{18}O\text{-}NO_3^-$ values (less than 31‰) were observed at the western edge of the Weddell Sea. The
air mass history of these samples indicates extensive interaction with sea ice covered ocean, which is known to enhance peroxy
radical production. The observed low $\delta^{18}O\text{-}NO_3^-$ is therefore attributed to increased exchange of NO with peroxy radicals,
which have a low $\delta^{18}O$, relative to ozone, which has a high $\delta^{18}O$. This study reveals that the mid- and high-latitude surface
ocean may serve as a more important $NO_x$ source than previously thought, and that the ice-covered surface ocean impacts the
reactive nitrogen budget as well as the oxidative capacity of the marine boundary layer.





# 1 Introduction

Atmospheric nitrate ($NO_3^-$), hereafter defined as gas phase nitric acid ($HNO_3$) and particulate $NO_3^-$ (p-$NO_3^-$), impacts air quality and climate by contributing to atmospheric particulate matter (Park & Kim, 2005), and influencing the Earths radiative heat budget (IPCC, 2013). It also plays a major role in the biogeochemical cycling of reactive nitrogen (Altieri et al., 2021). $NO_3^-$ aerosols originate from the oxidation of nitrogen oxides, collectively referred to as $NO_x$ ($NO_x$ = NO + $NO_2$). $NO_x$ cycling controls the chemical production of tropospheric ozone ($O_3$), a greenhouse gas and pollutant (Finlayson-Pitts & Pitts, 2000), which in turn contributes to the oxidising capacity of the atmosphere (Alexander & Mickley, 2015). Globally, fossil fuel combustion is the primary $NO_x$ source (van der A, et al., 2008), which far exceeds natural emissions such as biomass burning (Finlayson-Pitts & Pitts, 2000), soil processes (Davidson & Kingerlee, 1997) and lightning (Schumann & Huntrieser, 2007).

Due to its remoteness, the summertime Southern Ocean marine boundary layer (MBL) can be representative of preindustrial-like atmospheric conditions (Hamilton, et al., 2014). The chemical composition of the Southern Ocean MBL is characterised by low $NO_3^-$ concentrations (Virkkula, et al., 2006), representative of a background aerosol environment (i.e., minimal anthropogenic influence). Furthermore, the South Atlantic sector of the Southern Ocean is primarily influenced by natural $NO_x$ sources. During summer, high lightning activity over South America and southern Africa results in $NO_x$ production between approximately 40°S and the intertropical convergence zone (ITCZ) (Nesbitt, et al., 2000). As such, lightning is expected to be the dominant $NO_x$ source in the low latitude MBL (Schumann & Huntrieser, 2007; van der A, et al., 2008). Because of its pristine nature, the summertime Southern Ocean serves as a unique region in which to study atmospheric chemistry and is a useful preindustrial reference point for comparing the magnitude of anthropogenic aerosol impacts on climate (Haywood & Boucher, 2000; Hamilton, et al., 2014).

The atmospheric chemistry of the polar MBL at the high southern latitudes differs from that of the mid- and low-latitude MBL. During summer, high levels of photochemistry result in the emission of reactive gases from sea ice and snow cover in the Antarctic. As a result, highly elevated concentrations of hydrogen oxide radicals ($HO_x$ = OH + peroxy radicals), halogens, nitrous acid (HONO), and $NO_x$ have been observed during spring and summer in the polar regions (Brough et al., 2019). Furthermore, photochemical production of $NO_x$ within the surface snow of Antarctica and subsequent oxidation in the overlying atmosphere represents a significant $NO_3^-$ source to the Antarctic troposphere (Jones, et al., 2000; Jones, et al., 2001). $NO_3^-$ photolysis near the surface-air interface of ice crystals produces $NO_2$ (Grannas, et al., 2007; Jones, et al., 2000), which can be released to the firn (i.e., the intermediate stage of ice between snow and glacial ice) air and escape the snowpack to the overlying atmosphere (Erbland, et al., 2013; Shi, et al., 2015; Shi, et al., 2018). During winter, additional $NO_x$ sources to the Antarctic atmosphere may include long-range transported peroxyacetyl nitrates (PAN) and stratospheric inputs (Savarino, et al., 2007; Lee, et al., 2014; Walters, et al., 2019).

Emission of alkyl nitrates (a group of nitrogen gases collectively referred to as $RONO_2$) from the surface ocean have been recently proposed as a potential $NO_x$ source to the MBL in remote regions (Williams, et al., 2014; Fisher et al., 2018). Observations of elevated MBL alkyl nitrate concentrations suggest that a direct oceanic source exists in both the tropics (Atlas,





et al., 1993; Blake, et al., 2003), and the high-latitude Southern Ocean (Blake, et al., 1999; Jones, et al., 1999). Although the
exact mechanism remains unclear, experimental evidence suggests that oceanic $RONO_2$ production occurs via photochemical
processes involving the aqueous phase reaction of $RO_2$, derived from the photolysis of oceanic dissolved organic matter and
NO, derived from seawater nitrite photolysis (Dahl, et al., 2003; Dahl & Saltzman, 2008). Supersaturated $RONO_2$ conditions
in the surface ultimately drive a net flux from the ocean to the atmosphere (Chuck, et al., 2002; Dahl, et al., 2005). The
photolysis of emitted $RONO_2$ and subsequent OH oxidation in the overlying atmosphere leads to $NO_x$ formation (Fisher, et
al., 2018).
Current global atmospheric models suggest that oceanic $RONO_2$ represents a significant source of nitrogen (N) to the
Southern Ocean MBL, accounting for 20% to 60% of the reactive N pool at the high-latitudes (60°S to 90°S) (Fisher, et al.,
2018). However, only one small shipborne dataset with coincident ocean-atmosphere $RONO_2$ concentration measurements
exists to substantiate this notion (Hughes et al., 2008). Additionally, the $NO_x$ source from $RONO_2$ degradation dominates
relative to model defined primary $NO_x$ emission sources over the SO, which include shipping, aircraft and lightning (Fisher,
et al., 2018). However, the lack of seawater observations available to constrain Southern Ocean $RONO_2$ distributions hamper
the validation of model fluxes. Better understanding of the Southern Ocean $RONO_2$ source is required to improve simulations
and accurately evaluate its contribution to the Southern Ocean MBL $NO_x$ budget.
**1.1 Natural abundance isotopes of atmospheric nitrate**
Measurements of the oxygen (O) and N stable isotope ratios of atmospheric $NO_3^-$ can be used to constrain $NO_x$ sources, NO-
$NO_2$ cycling, and $NO_x$ to $NO_3^-$ oxidation pathways, which are critical to our understanding of the reactive N budget in the
atmosphere. This technique has been applied in both polluted (Elliot, et al., 2007; Zong, et al., 2017), open ocean (Hastings,
et al., 2003; Morin, et al., 2009; Kamezaki, et al., 2019; Gobel, et al., 2013; Altieri, et al., 2013), and polar environments
(Morin et al., 2009; Walters et al., 2019). Stable isotope ratios are reported as a ratio of the heavy to light isotopologues of a
sample relative to the constant isotopic ratio of a reference standard, using delta (δ) notation in units of "per mil" (‰) following
Eq. (1):
$$\delta = \left( \left( R_{sample}/R_{standard} \right) - 1 \right) \times 1000 \qquad (1)$$
where R represents the ratio of $^{15}N/^{14}N$ or $^{18}O/^{16}O$ in the sample and in the reference standard, respectively. The reference for
O is Vienna Standard Mean Ocean Water (VSMOW) and for N is atmospheric $N_2$ (Bolhke, et al., 2003).
When $NO_x$ is converted to $NO_3^-$, the N atom is conserved. As such, it is generally expected that the N stable isotope ratio
of atmospheric $NO_3^-$ ($\delta^{15}N$-$NO_3^-$) reflects the $\delta^{15}N$ of the source $NO_x$, (Kendall, et al., 2007) plus any isotopic fractionation
associated with NO/$NO_2$ cycling or $NO_x$ to $NO_3^-$ conversion. For example, the $\delta^{15}N$ of lightning generated $NO_x$ is close to 0‰
(Hoering, 1957) and is distinct from stratospheric and snowpack $NO_x$. Savarino et al., (2007) used the degree of $N_2O$
destruction in the stratosphere and the associated isotopic fractionation to derive an Antarctic stratospheric $\delta^{15}N$-$NO_x$ source
signature of 19‰ ± 3‰ (Savarino, et al., 2007). In contrast, snow emitted $NO_x$ typically has a very low $\delta^{15}N$ signature due to



the large fractionation ($^{15}\varepsilon$) of $\sim$ - 48‰ (Berhanu et al., 2014 and 2015) associated with $NO_3^-$ photolysis in the snowpack,
where $^{15}\varepsilon$ of a reaction is the ratio of the rates with which the two isotopes of N are converted from reactant to product. If
equilibrium isotope fractionation during $NO/NO_2$ cycling occurs, it results in the $^{15}N$ enrichment of $NO_2$ such that the $NO_3^-$
formed from this $NO_2$ will have a higher $\delta^{15}N$-$NO_3^-$ than the initial $NO_x$ source (Freyer, et al., 1993; Walters, et al., 2016).
Equilibrium isotope fractionation during the transformation of $NO_x$ to $NO_3^-$ also results in higher $\delta^{15}N$-$NO_3^-$ compared to the
original $NO_x$ source (Walters & Michalski, 2015).
In contrast to N, the O stable isotope ratio of atmospheric $NO_3^-$ ($\delta^{18}O$-$NO_3^-$) is reflective of the oxidants involved in $NO_x$
cycling prior to $NO_3^-$ formation, as well as the dominant $NO_3^-$ formation pathway (Hastings, et al., 2003; Michalski, et al.,
2003; Alexander, et al., 2020). The O atoms of $NO_x$ are rapidly exchanged with oxidising agents in the atmosphere to produce
$NO_3^-$. Tropospheric $NO_x$ recycles rapidly with $O_3$ following the equations below:
$NO + O_3 \rightarrow NO_2 + O_2$ (R1)
$NO_2 + O_2 + hv \rightarrow NO + O_3$ (R2)
The oxidation of NO to $NO_2$ requires an atmospheric oxidant, typically $O_3$ throughout most of the troposphere (R1), while the
breakdown of $NO_2$ back to NO is photolytic and requires light (R2). Therefore, under nighttime/dark conditions (R2) shuts
down and $NO_x$ is comprised almost entirely of $NO_2$.
The dominant daytime sink for $NO_x$ is the oxidation of $NO_2$ by OH, which produces nitric acid ($HNO_3$) via (R3), where M is
a non-reacting molecule.
$NO_2 + OH + M \rightarrow HNO_3 + M$ (R3)
Under nighttime/dark conditions, the photolytic production of OH cannot occur and $NO_2$ is oxidised by $O_3$ (R4). $HNO_3$ is
ultimately formed via the hydrolysis of dinitrogen pentoxide ($N_2O_5$), following the reactions (R5) and (R6) below:
$NO_2 + O_3 \rightarrow NO_3 + O_2$ (R4)
$NO_3 + NO_2 + M \rightleftharpoons N_2O_5 \text{ (g)} + M$ (R5)
$N_2O_5 \text{ (g)} + H_2O \text{ (l)} + surface \rightarrow 2HNO_3 \text{ (aq)}$ (R6)
$NO_3$ can also react with hydrocarbons (HC) (e.g., dimethylsulphide (DMS)) to form $HNO_3$ following reaction (R7) below:
$NO_3 + HC/DMS \rightarrow HNO_3 + products$ (R7)
Lastly, in regions with elevated halogen concentrations, $NO_2$ can be oxidised by reactive halogens, for example bromine oxide
(BrO), to form $HNO_3$ following (R8) and (R9) below:
$NO_2 + BrO \rightarrow BrONO_2$ (R8)
$BrONO_2 + H_2O + surface \rightarrow HNO_3 + HOBr$ (R9)
Typically, aerosol $\delta^{18}O$-$NO_3^-$ is interpreted as being determined by the dominant $NO_x$ oxidation pathways, (R3) versus
(R4) to (R9). If some combination of R4-R9 occurs, then $O_3$ is the main oxidant, whereas during (R3), one of the O atoms
originates from OH. The OH radical exchanges with $H_2O$ vapor in the troposphere, therefore the $\delta^{18}O$ of OH is a function of
the $\delta^{18}O$ of $H_2O$ vapour, which generally ranges from -27.5‰ to 0‰ in the subtropics and over the Southern Ocean (Michalski,
et al., 2012; Guilpart, et al., 2017; Dar, et al., 2020), and equilibrium isotope exchange between OH and $H_2O$ (Walters &





Michalski, 2016). In contrast the $\delta^{18}O$ of tropospheric $O_3$ is much higher, the most recent estimate being 114.8±10.4‰ (Vicars
& Savarino, 2014). Therefore, a higher $\delta^{18}O$ for atmospheric $NO_3^-$ reflects the increased influence of $O_3$ on $NO_x$ to $NO_3^-$
conversion (R4-R9), and the $\delta^{18}O\text{-}NO_3^-$ is lower when (R3) is favoured, due to the lack of exchange of O atoms with $O_3$
(Hastings, et al., 2003; Fang, et al., 2011; Altieri, et al., 2013).
Here, we present the concentration and isotopic composition of coarse mode (> 1 μm) atmospheric $NO_3^-$ collected in the
MBL of the Southern Ocean between Cape Town, South Africa and coastal Antarctica, as well as across the Weddell Sea gyre,
during summer. Using air mass back trajectories, surface ocean nitrite measurements, and the aerosol $\delta^{15}N\text{-}$ and $\delta^{18}O\text{-}NO_3^-$,
we address 1) the major $NO_x$ sources as well as the main oxidants in $NO/NO_2$ cycling and $NO_x$ to $NO_3^-$ conversion across a
large latitudinal transect of the Atlantic Southern Ocean and within the Weddell Sea gyre, and 2) the influence of sea-ice and
snowpack emissions on $NO_x/NO_3^-$ chemistry in the high-latitude MBL.
**2) Methods**
**2.1) Sample collection**
Samples were collected on board the Research Vessel (R/V) *SA Agulhas II* during one cruise subdivided into three legs. Leg
one refers to the voyage south from Cape Town (33.9°S, 18.4°E) to Penguin Bukta (71.4°S, 2.5°W) in early summer (7
December 2018 to 19 December 2018) as part of the South African National Antarctic Expedition's annual relief voyage
(SANAE 58). Leg two is the Weddell Sea Expedition (WSE) from 4 January 2019 to 21 February 2019. The WSE refers to
the voyage west from Penguin Bukta to the northern edge of the Weddell Sea gyre to Larsen C ice shelf, followed by a detour
to King George Island before returning to the Weddell Sea and sailing back to Penguin Bukta. Leg three refers to the SANAE
return voyage north from Penguin Bukta to Cape Town in late summer (27 February 2019 to 15 March 2019). From here
on legs one, two and three will be referred to as early summer, the Weddell Sea, and late summer, respectively.
Size-segregated atmospheric aerosols were collected on the ninth floor above the bridge (approximately 20 m above sea
level), using a high-volume air sampler (HV-AS; Tisch Environmental). Air was pumped at an average flow rate of 0.82 m$^3$
min$^{-1}$ though a five-stage cascade impactor (TE-235; Tisch Environmental), loaded with combusted (400°C for 4 hours) glass
fibre filters. Aerosol nitrate in the MBL is predominantly present in the coarse mode (> 1 μm), therefore only filter stages 1
through 4 were analysed, where the aerodynamical diameter of particles collected are as follows: stage 1 (> 7 μm); stage 2 (3
to 7 μm); stage 3 (1.5 to 3 μm) and stage 4 (1 to 1.5 μm).
A sector collector was used to restrict HV-AS activity to avoid contamination from ship stack emissions (Campbell
Scientific Africa). The HV-AS only began operating if the wind was blowing at an angle less than 75° or greater than 180°
from the bow of the ship for a minimum of ten minutes at a speed of at least 1 m s$^{-1}$. Filters were removed from the cascade
impactor inside a laminar flow cabinet (Air Science), placed in individual zip-sealed plastic bags and stored at -20°C until
analysis.
Given that the MBL of the Southern Ocean is characterised by low atmospheric $NO_3^-$ concentrations, an attempt was made
to ensure that at least 24 hours of in-sector sampling had passed before filters were removed from the cascade impactor.
However, this was not always possible as on occasion the filters had to be removed early to avoid contamination due to unusual
ship manoeuvres resulting in stagnant conditions. Therefore, sampling times ranged between 13 and 88 hours across the three
legs. The details of each cruise leg can be found in the supplemental information (Table S1).
During the research voyage, a field blank was collected by fitting the cascade impactor with a set of filters and walking the
cascade impactor from the laboratory to the HV-AS in the same way that atmospheric samples were deployed. The cascade
impactor was placed into the HV-AS and then immediately removed without the HV-AS turning on, after which the filters
were removed from the cascade impactor and stored in the same manner as the atmospheric samples. All chemical analyses
performed on samples were also performed on the field blanks to assess possible contamination during filter deployment or
sample handling.
**2.2) Sample analysis**
Filter stages 1 to 4 were extracted using ultra-clean deionised water (DI) under a laminar flow cabinet (Air Science). The
extraction ratio was approximately 30 $cm^2$ of filter in 25 mL of DI. Extracts were immediately sonicated for one hour and then
stored at 4°C for at least 12 hours. Thereafter, extracts were filtered (0.2 μm) using an acid washed syringe into a clean 30 mL
HDPE bottle and stored at -20°C until analysis (Baker, et al., 2010).
Aerosol nitrate concentrations ([$NO_3^-$]) were determined using a Thermo Scientific Dionex Aquion Ion Chromatography
(IC) system equipped with an autosampler. The anion IC contained an AG22 RFIC 4 x 50 mm guard column and AG22 RFIC
4 x 250 mm analytical column. A six-point standard curve was run on each day of analysis (Dionex Seven Anion-II Standard)
and an $R^2$ value > 0.999 was required for sample analysis to proceed. Final [$NO_3^-$] were corrected by subtracting the field
blanks, which had an average of 484.7 nmol $NO_3^-$ per filter deployment. The pooled standard deviation (Sp) of four repeated
sample measurements for [$NO_3^-$] was 0.3 μmol $L^{-1}$. A subset of aerosol samples was analysed for [$NO_3^-$] using a Lachat
QuikChem® flow injection autoanalyzer (precision of ± 0.8 μmol $L^{-1}$). For the subset of samples analysed using both
instruments, the average [$NO_3^-$] measured using the Lachat QuikChem® flow injection autoanalyzer and the IC system is
reported.
Nitrogen and oxygen isotopic ratios were measured using the denitrifier method (Sigman, et al., 2001 and Casciotti, et al.,
2002). To determine the $^{15}N/^{14}N$ and $^{18}O/^{16}O$ of $NO_3^-$, a natural strain of denitrifying bacteria, *Pseudomonas aureofaciens*, that
lack the terminal nitrous oxide ($N_2O$) reductase enzyme were used to convert aqueous $NO_3^-$ quantitatively to $N_2O$ gas. The
product $N_2O$ was analysed by continuous flow isotope ratio mass spectrometry (IRMS) using a Delta V Advantage IRMS
interfaced with an online $N_2O$ extraction and purification system. Individual analyses were referenced to injections of $N_2O$
from a pure gas cylinder and then standardized through comparison to the international reference materials of IAEA-N3 and
USGS34 for $\delta^{15}N$-$NO_3^-$, and IAEA-N3, USGS34 and USGS35 for $\delta^{18}O$-$NO_3^-$ (Table S2) (Bohlke et al., 2003). The $^{15}N/^{14}N$ of
samples was corrected for the contribution of $^{17}O$ to the peak at mass 45 using an average reported $\Delta^{17}O$ value of 26‰ from





atmospheric nitrate collected in the Weddell Sea (Morin, et al., 2009). The pooled standard deviation for all measurements of
IAEA-N3 and USGS34 for $\delta^{15}N$-$NO_3^-$, and IAEA-N3, USGS34 and USGS35 for $\delta^{18}O$-$NO_3^-$ are reported (Table S2). All
samples were measured in triplicate in separate batch analyses. The pooled standard deviation from all replicate analyses of
samples was 0.25‰ for $\delta^{15}N$-$NO_3^-$ and 0.64‰ for $\delta^{18}O$-$NO_3^-$. The average $\delta^{15}N$-$NO_3^-$ and $\delta^{18}O$-$NO_3^-$ computed for each filter
deployment was weighted by the $[NO_3^-]$ observed for each stage and error was propagated according to standard statistical
practises (Table S3).
Seawater samples were collected in triplicate every two hours from the ships underway system (position at depth ± 5 m)
for the analysis of surface ocean nitrite concentrations ($[NO_2^-]$). $[NO_2^-]$ was analysed using the colorimetric method of Grasshof
et al. (1983) using a Thermo Scientific Genesys 30 visible spectrophotometer (detection limit of 0.05 µmol $L^{-1}$).
**2.6) Air mass back trajectory analysis**
To determine the air mass source region for each aerosol sample, air mass back trajectories (AMBTs) were computed for each
hour in which the HV-AS was operational for at least 45 minutes of that hour. Given that the ship was moving, a different
date, time and starting location was used to compute each AMBT. An altitude of 20 m was chosen to match the height of the
HV-AS above sea level and 72-hour AMBTs were computed to account for the lifetime of $NO_3^-$ in the atmosphere. All AMBTs
were computed with NOAA's Hybrid Single-Particle Lagrangian Integrated Trajectory model (HYSPLIT v 4), using NCEP
Global Data Assimilation System (GDAS) output, which can be accessed at http://www.arl.noaa.gov/ready/hysplit4.html
(NOAA Air Resources Laboratory, Silver Spring, Maryland) (Stein, et al., 2015; Rolph, 2016).
**3) Results**
The coarse mode (> 1 µm in diameter) aerosol $[NO_3^-]$ computed by summing the $[NO_3^-]$ of stages 1 through 4, ranged from
22.3 to 374.2 ng $m^{-3}$ (Fig. 1A and Table 1). The mass-weighted $\delta^{15}N$ of coarse mode aerosol $NO_3^-$ ranged from -43.1‰ to -
2.7‰ (Figs. 1B, 2 and Table 1). The highest nitrate concentrations occurred between 34˚S and 45˚S, and then decreased with
increasing latitude. Similarly, higher values characterized $\delta^{15}N$-$NO_3^-$ between 34˚S and 45˚S (-4.5 ± 1.6‰), and then decreased
with increasing latitude.  At the high-latitudes (south of 60˚S), median values of 41.7 ng $m^{-3}$ and -22.2‰ were observed for
nitrate concentration and $\delta^{15}N$, respectively. Coincident mass-weighted $\delta^{18}O$-$NO_3^-$ values ranged from 16.5‰ to 70‰ (Figs.
1C, 3 and Table 1). No latitudinal trend in $\delta^{18}O$-$NO_3^-$ was apparent, although distinctly low $\delta^{18}O$-$NO_3^-$ values were observed
in the Weddell Sea, as discussed in section 4.3 below. The difference between $\delta^{18}O$-$NO_3^-$ observed in the Weddell Sea (during
January to February) and $\delta^{18}O$-$NO_3^-$ observed at corresponding latitudes during the early and late summer transects is
statistically significant (p-value < 0.05). The early and late summer cruise transects were similar spatially in that both took
place along the same hydrographic line (i.e., the Good Hope line), apart from the deviation to South Georgia during late
summer (Fig. 2A & B). Even though the early and late summer cruise transects occurred in December and March, respectively,





there is no statistically significant difference in [NO$_3^-$], $\delta^{15}$N-NO$_3^-$ or $\delta^{18}$O-NO$_3^-$ between them (p-value > 0.05 in all cases).
Therefore, the early and late summer legs are discussed together and collectively referred to as the latitudinal transect.

Table 1: The average (Avg), standard deviation (SD) and range of total coarse-mode (> 1µm) atmospheric nitrate
concentration ([NO$_3^-$]; ng m$^{-3}$) and the mass weighted average N and O isotopic composition of coarse mode nitrate ($\delta^{15}$N-
NO$_3^-$ and $\delta^{18}$O-NO$_3^-$; ‰) are shown. Cruise legs are denoted as follows: early summer (ES), Weddell Sea (WS) and late
summer (LS).

| Leg | [NO$_3^-$] (ng m$^{-3}$) | | $\delta^{15}$N-NO$_3^-$ (‰ vs. N2) | | $\delta^{18}$O-NO$_3^-$ (‰ vs. VSMOW) | |
|---|---|---|---|---|---|---|
| | Avg (SD) | Range | Avg (SD) | Range | Avg (SD) | Range |
| ES | 139 (112.8) | 31.9 to 374.2 | -19.5 (16.4) | -43.1 to -2.7 | 47.1 (17.7) | 16.5 to 70.0 |
| WS | 46.7 (19.5) | 22.3 to 94.8 | -22.7 (7.1) | -38.1 to -11.6 | 38.3 (12.8) | 18.8 to 60.3 |
| LS | 94.0 (95.5) | 22.3 to 282.5 | -15.3 (8.4) | -25.6 to -4.6 | 51.1 (5.5) | 44.9 to 58.9 |



Fig. 1. The average (± 1 SD) coarse mode (> 1 μm) nitrate concentration [NO₃⁻] (ng m⁻³; A), and the weighted average (± 1 SD) δ¹⁵N (B) and δ¹⁸O (C) of atmospheric nitrate (δ¹⁵N-NO₃⁻ (‰ vs. N₂) and δ¹⁸O-NO₃⁻ (‰ vs. V-SMOW), respectively), as a function of latitude (°S). Early and late summer latitudinal transects are denoted by the red triangles and green squares, respectively. Weddell Sea samples are denoted by blue circles. Where error bars (± 1 SD) are not visible, the standard deviation is smaller than the size of the marker.



Fig. 2. 72-hour AMBT's (grey lines) computed for each hour of the voyage when the HV-AS was operational for more than 45 minutes of the hour during early summer (A), late summer (B), and in the Weddell Sea (C). The colour bar represents the weighted average $\delta^{15}N$ of coarse mode (> 1 μm) atmospheric nitrate ($\delta^{15}N$-$NO_3^-$).



Fig. 3. 72-hour AMBT's (grey lines) computed for each hour of the voyage when the HV-AS was operational for more than 45 minutes of the hour during early summer (A), late summer (B), and in the Weddell Sea (C). The colour bar represents the weighted average $\delta^{18}O$ of coarse mode (> 1 μm) atmospheric nitrate ($\delta^{18}O\text{-}NO_3^-$).



**4) Discussion**
The sum of our observations reveals a latitudinal gradient in atmospheric $NO_3^-$ concentration and $\delta^{15}N$-$NO_3^-$, which we
hypothesize may be attributed to the varying contribution of the dominant $NO_x$ sources present between Cape Town and coastal
Antarctica. In contrast, $\delta^{18}O$-$NO_3^-$ depicts no latitudinal trend, however, very low $\delta^{18}O$-$NO_3^-$ values are observed in the Weddell
Sea, which we hypothesize may be attributed to the influence of sea ice emissions on $NO_x$ cycling. Below, we first discuss the
extent to which anthropogenic $NO_x$ sources may influence the observed atmospheric $NO_3^-$ concentrations and $\delta^{15}N$ signatures.
Then we discuss the dominant $NO_x$ sources to low, mid and high latitude Southern Ocean MBL $NO_3^-$, determined in part from
72-hour AMBT's, as well as the role of various oxidants in $NO/NO_2$ cycling and $NO_2$ oxidation.
**4.1) Minimal influence of anthropogenic $NO_x$ sources**
Aerosol $NO_3^-$ concentrations were low (< 100 ng m$^{-3}$; Fig. 1A) for most air masses sampled along the latitudinal transect
and in the Weddell Sea, consistent with the expectation of minimal influence from anthropogenic $NO_x$ sources. Interestingly,
$NO_3^-$ concentrations were higher ($\pm$ 300 ng m$^{-3}$; Fig. 1A) in samples collected near the South African coast at the beginning of
the latitudinal transect (i.e., above 43°S). However, 72-hour AMBTs computed for all latitudinal transect samples indicate that
sampled air masses originated from over the South Atlantic sector of the Southern Ocean (Fig. 2A and 2B), with no continental
influence and limited opportunity for direct anthropogenic $NO_x$ emissions to contribute to aerosol $NO_3^-$, assuming $NO_3^-$ has a
lifetime of 72 hours (Alexander, et al., 2020). As such, the higher atmospheric $NO_3^-$ concentrations observed near South Africa
are best explained by greater lightning $NO_x$ production, which generally occurs between 40°S and the ITCZ during summer
(Nesbitt, et al., 2000; van der A, et al., 2008).

**4.2) Interpretation of natural $NO_x$ sources using the N isotopic composition of atmospheric $NO_3^-$**
Aerosol $\delta^{15}N$-$NO_3^-$ ranged from -2.7‰ for low-latitude air masses to -43.1‰ for high-latitude air masses (including those
sampled in the Weddell Sea; Fig. 1B). As discussed in section 1.1, the $\delta^{15}N$-$NO_3^-$ reflects the $\delta^{15}N$ of the source $NO_x$ plus any
isotopic fractionation imparted from $NO/NO_2$ cycling or $NO_x$ to $NO_3^-$ conversion. Similar to previous studies, we surmise that
$NO_x$ equilibrium fractionation is unlikely to be relevant in our system, as $NO_x$ concentrations are significantly lower than $O_3$
concentrations (Elliott, et al., 2007; Morin, et al., 2009; Walters, et al., 2016; Park, et al., 2018). Typical $O_3$ concentrations
observed at coastal sites in Antarctica are on the order of 20 ppbv (Nadzir, et al., 2018), whereas the sum of NO and $NO_2$ rarely
exceeds 40 pptv (Jones, et al., 2000; Weller, et al., 2002; Bauguitte, et al., 2012). Under these conditions $NO_x$ isotopic exchange
occurs at a much slower rate than (R1) and (R2), such that little to no equilibrium isotope fractionation is expressed and the
$\delta^{15}N$ of the $NO_3^-$ should reflect the $\delta^{15}N$ of the $NO_x$ source (Walters, et al., 2016). Additionally, equilibrium isotope effects are
temperature dependent (increasing with decreasing temperature) and here ambient temperatures decline with increasing
latitude. Therefore, if equilibrium isotope fractionation were occurring during $NO$-$NO_2$ cycling and/or $NO_x$ to $NO_3^-$ conversion,





one would expect $\delta^{15}N\text{-}NO_3^-$ to increase with latitude, as both fractionation processes produce $NO_3^-$ with a $\delta^{15}N$ signature
higher than the source $NO_x$. However, the opposite trend is observed here whereby $\delta^{15}N\text{-}NO_3^-$ decreases with increasing
latitude (Fig. 1B). Therefore, we discount the hypothesis that equilibrium isotope effects can explain the latitudinal gradient
in $\delta^{15}N\text{-}NO_3^-$.
$NO_3^-$ in the Antarctic troposphere may also derive from stratospheric denitrification, whereby $HNO_3$ is injected into the
troposphere from the stratosphere via the subsidence and penetration of polar stratospheric clouds (PSC). However, this
phenomenon typically occurs in winter when the tropospheric barrier is weak and the lower stratosphere is cold enough for
PSC formation (Savarino, et al., 2007; Walters, et al., 2019). Furthermore, $\delta^{15}N\text{-}NO_3^-$ originating from stratospheric inputs is
estimated to be 19‰ ± 3‰ (Savarino, et al., 2007), a value substantially greater than the atmospheric $\delta^{15}N\text{-}NO_3^-$ observed here
for high-latitude air masses; thus, we discount a direct influence from stratospheric $NO_x$. We propose that the observed
variation in atmospheric $\delta^{15}N\text{-}NO_3^-$ across the Southern Ocean is therefore best explained by the changing contribution of three
dominant $NO_x$ sources: lightning, surface ocean alkyl nitrate emissions, and photochemical production on snow and ice,
determined using AMBT analyses and typical $NO_x$ source signatures where possible, as discussed below.

**4.2.1) High-latitudes: Photochemical $NO_x$ source**

Aerosol $\delta^{15}N\text{-}NO_3^-$ was very low in air masses from the southern high-latitudes, including in the Weddell Sea (average of
-24.3‰; Figs. 1B & 2). The latitudinal gradient in lightning suggests that $NO_x$ production via this mechanism is greatly reduced
at high-latitudes (Savarino, et al., 2007). Similar to other studies in the region (Savarino, et al., 2007; Morin, et al., 2009), we
suggest that photochemical $NO_x$ production on snow or ice accounts for the low aerosol $\delta^{15}N\text{-}NO_3^-$ in high-latitude air masses,
where high-latitude air mass samples are defined as those exposed to the Antarctic continent or the surrounding sea ice (with
sea ice concentration being at least 50%) (Fig. 4, red). Antarctic estimates for isotopic fractionation associated with snow $NO_3^-$
photolysis during summer range from -47.9‰ to -55.8‰ for laboratory and field experiments, respectively (Berhanu, et al.,
2014, 2015), resulting in the emission of low $\delta^{15}N$ $NO_x$ to the overlying atmosphere (Savarino, et al., 2007; Morin, et al., 2009;
Shi, et al., 2018; Walters, et al., 2019). Therefore, $NO_3^-$ photolysis explains the very low $\delta^{15}N\text{-}NO_3^-$ observed in high-latitude
air masses in early and late summer that crossed snow-covered continental ice or sea ice before being sampled (Figs. 2 and 4).
During early summer, air masses spent significantly more time over the snow-covered continent compared to late summer
(Figs. 2A & B) and the sea ice extent was greater in early summer compared to late summer (Fig. 4). Combined, these dynamics
resulted in a much lower $\delta^{15}N\text{-}NO_3^-$ for high-latitude air masses during early summer compared to late summer (minimum
value of -43.1‰ vs -25.6‰). Similarly low MBL $\delta^{15}N\text{-}NO_3^-$ values (< -30‰) were recently observed for the southern high
latitudes of the Indian ocean (Shi, et al., 2021). Our data are also consistent with previous year-round studies of atmospheric
$NO_3^-$ at coastal Antarctica (Savarino, et al., 2007) and the South Pole (Walters, et al., 2019), where $\delta^{15}N\text{-}NO_3^-$ was reported
to range from -46.9‰ to 10.8‰ and from -60.8‰ to 10.5‰, respectively. Both studies observed a seasonal cycle in $\delta^{15}N\text{-}$
$NO_3^-$ whereby the lowest values occurred during sunlit periods (i.e., summer) due to snowpack $NO_x$ emissions and the highest
values occurred during dark periods (i.e., winter) due to stratospheric inputs (Savarino, et al., 2007; Walters, et al., 2019).

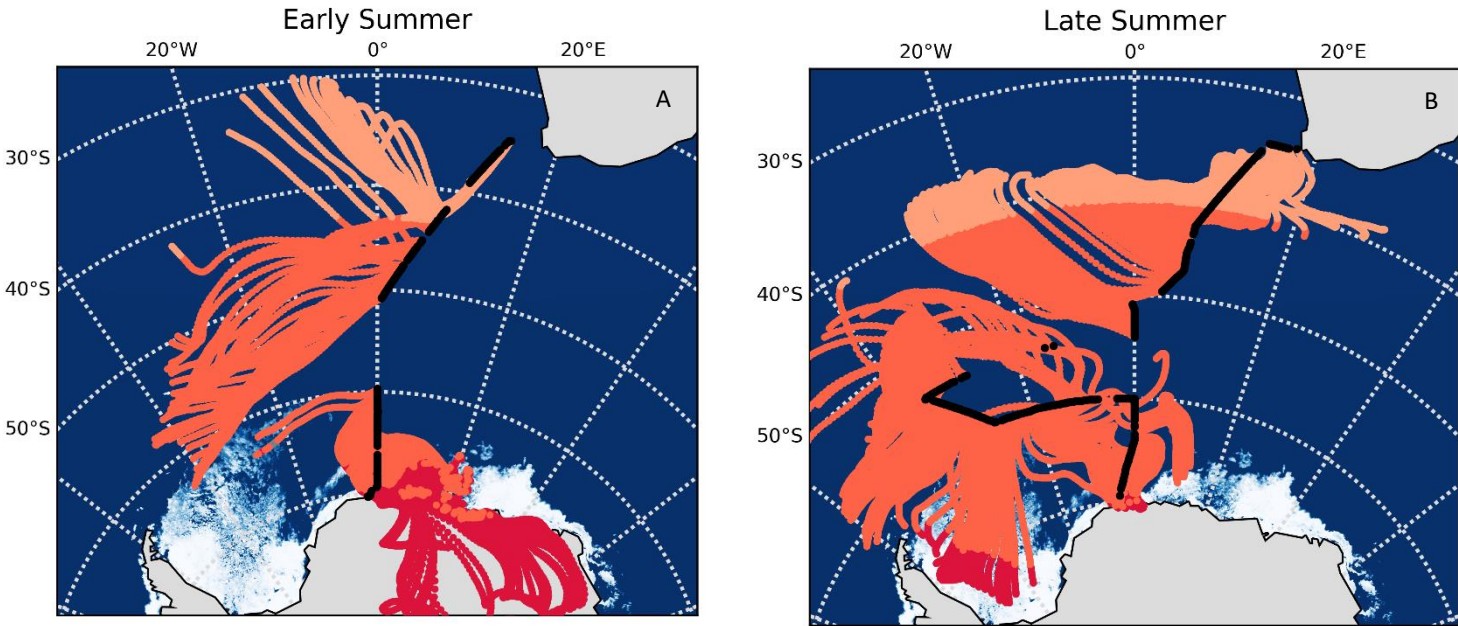

Fig. 4. 72-hour AMBT's computed for each hour of the voyage during early (A) and late (B) summer, when the HV-AS was operational for more than 45 minutes of the hour. Light orange, dark orange and red AMBT's represent time spent over the low, mid- and high latitude SO, respectively. The white represents the location of the sea-ice at the southernmost extent of each transect. Satellite derived sea ice concentration was obtained from the AMSR2 ASI programme.

**4.2.2) Low- to Mid-latitudes: Oceanic NO$_x$ source**

At the northern extent of our transects, the low-latitude aerosol samples, defined as those originating from anywhere north of 43°S in early summer and 41°S in late summer (Fig. 4, light orange), had the highest average $\delta^{15}$N-NO$_3^-$ signature (-4.9 ± 1.5‰; n = 5). These values can be attributed to lightning-generated NO$_x$, which has a $\delta^{15}$N signature close to 0‰ (Hoering, 1957). Lightning activity at the low latitudes is also consistent with the higher atmospheric [NO$_3^-$] observed (Fig. 1A). An average atmospheric $\delta^{15}$N-NO$_3^-$ signature of -4‰ was previously reported for the low latitude Atlantic Ocean, between 45°S and 45°N, and similarly attributed to a combination of natural NO$_x$ sources including lightning (Morin, et al., 2009).

Aerosol samples across the mid-latitudes had an average $\delta^{15}$N-NO$_3^-$ of -13.2‰ (Figs. 1B & 2). Mid-latitude air masses are defined as those originating from anywhere south of 43°S in early summer and south of 41°S in late summer that made no contact with Antarctica or any surrounding sea ice (Fig. 4, dark orange), therefore these samples were unlikely to be influenced

by snow emitted NO$_x$ with its light isotopic signature. The beginning of the mid-latitude zone (i.e., end of the low-latitude
zone) was defined by the presence of non-zero sea surface nitrite concentrations in early and late summer (Fig. 5). However,
the observed aerosol δ$^{15}$N-NO$_3^-$ was too low (-14.5‰ to -11.2‰) to be explained solely by lightning generated NO$_x$. In the
absence of any signature of anthropogenic NO$_x$ emissions (see Sect. 4.1), we argue that the dominant NO$_x$ source for the mid-
latitude samples originates from seawater.

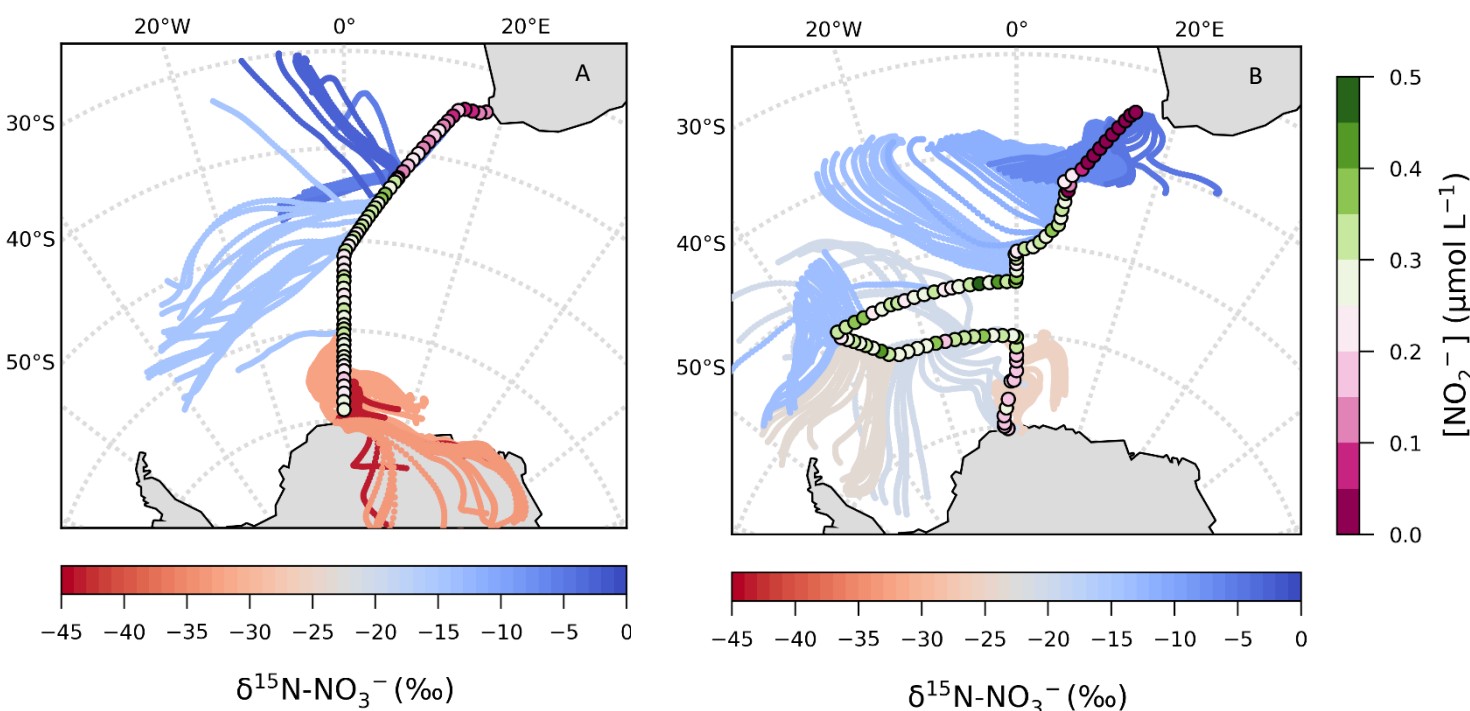

Fig. 5. 72-hour AMBT's computed for each hour of the voyage during early (A) and late (B) summer, when the HV-AS
was operational for more than 45 minutes of the hour. AMBT's are colour coded by the weighted average δ$^{15}$N of atmospheric
nitrate (δ$^{15}$N-NO$_3^-$), represented by the horizontal colour bar. Over-layered are the surface ocean nitrite concentrations (circles;
[NO$_2^-$]; μmol L$^{-1}$), measured along each transect and represented by the vertical colour bar.

As mentioned in section 1, the most likely mechanism for an oceanic NO$_x$ source is via the photolysis of surface ocean
derived RONO$_2$ in the MBL. NO derived from seawater nitrite is thought to limit RONO$_2$ production (dahl & Saltzman 2008;
Dahl et al., 2012), such that non-zero nitrite concentrations are required for RONO$_2$ production to occur. Here, surface ocean
nitrite concentrations were relatively high, in particular from ~ 41°S to 50°S (Fig. 5).  Furthermore, the latitudinal extent of





mid-latitude air masses with low $\delta^{15}$N-NO$_3^-$ signatures corresponds well with the same latitudinal extent in which non-zero
surface ocean nitrite concentrations occurred (Figs. 4 and 5). As such, we suggest that in this region oceanic RONO$_2$ emission
is the main source to the Southern Ocean MBL, ultimately resulting in the low $\delta^{15}$N-NO$_3^-$ values observed for mid-latitude air
masses.
No estimates exist for the $\delta^{15}$N of oceanic RONO$_2$, however RONO$_2$ photolysis may yield isotopically light NO$_x$ given that
NO$_3^-$ photolysis produces low $\delta^{15}$N products (e.g., Frey et al., 2009). Therefore, once oxidised in the overlying atmosphere,
NO$_x$ derived from oceanic RONO$_2$ photolysis may form atmospheric NO$_3^-$ with a low $\delta^{15}$N signature. Aerosol $\delta^{15}$N-NO$_3^-$
values have been observed to range from -14.1‰ to -7.3‰ in the eastern equatorial Pacific (Kamezaki, et al., 2019) and from
-6‰ to ∼0‰ (average = -3.4‰) in the western equatorial Pacific (Shi, et al., 2021). Observed $\delta^{15}$N-NO$_3^-$ is higher in the
western compared the eastern equatorial Pacific, which could be attributed to the proximity of the western equatorial Pacific
to continental/anthropogenic NOx sources, resulting in NO$_3^-$ having a higher $\delta^{15}$N signature. The low average $\delta^{15}$N-NO$_3^-$
observed for the mid-latitude air masses of the Southern Ocean MBL sampled in the present study (-14.5‰ to -11.2‰), are
remarkably similar to those observed in the eastern equatorial Pacific (Kamazaki, et al., 2019). Kamezaki et al., (2019) also
concluded that such low $\delta^{15}$N-NO$_3^-$ values cannot be explained solely by lightning NO$_x$ and given the lack of considerable
influence from any continental NO$_x$ sources, they invoked the contribution of oceanic N emissions in the form of ammonia
(NH$_3$) and/or RONO$_2$. However, NH$_3$ flux data for the summertime Atlantic Southern Ocean derived from in situ
ocean/atmosphere observations suggest that the ocean in this region is a net sink of NH$_3$ (Altieri, et al., 2021).
The latitudinal extent of our sampling campaign enabled us to estimate a range of likely values for the N isotopic
composition of NO$_3^-$ derived from oceanic RONO$_2$. We split the latitudinal transect into three regions, each characterised by
the dominance of a different natural source of NO$_3^-$ i.e., lightning NO$_x$ at the low-latitudes, oceanic RONO$_2$ emissions at the
mid-latitudes and snowpack emissions at the high-latitudes. Assuming that the dominant natural source of NO$_3^-$ is the only
source relevant in each latitudinal zone, we estimate the contribution of each source to total NO$_3^-$ formation by ascertaining
the amount of time air masses spent in each zone. We further assume that atmospheric $\delta^{15}$N-NO$_3^-$ reflects at most a combination
of two sources based on the AMBTs of each sample, either lightning NO$_x$ and oceanic RONO$_2$ emissions near South Africa,
or oceanic RONO$_2$ emissions and snowpack NO$_x$ emissions near Antarctic (Fig. 4 and Table S4). Using a two-end member
mixing model the $\delta^{15}$N signature of the source NO$_3^-$ derived from mid-latitude Southern Ocean RONO$_2$ emissions was
calculated for all samples where air masses from the mid-latitude region contributed at least 10% (Table S4). This 10%
threshold was chosen as the isotopic endmember of oceanic RONO$_2$ is harder to determine with confidence when its
contribution to total NO$_3^-$ is less than 10%. For example, the AMBT's for sample ES 4 spent 3% of the time in the low-latitude
zone and 97% in the mid-latitude zone. Therefore, assuming lightning NO$_x$ has a $\delta^{15}$N signature of 0‰ and the measured $\delta^{15}$N-
NO$_3^-$ for ES 4 is -14.5‰, we calculate the $\delta^{15}$N signature of the RONO$_2$-derived NO$_3^-$ to be -14.9‰.
Using this approach for each filter deployment along the latitudinal transect, an average $\delta^{15}$N-NO$_3^-$ from oceanic RONO$_2$
emissions of -22.0 ± 7.5‰ was estimated. Furthermore, the contribution of RONO$_2$ emissions can explain the lowering of
$\delta^{15}$N from 0‰ for the low-latitude air mass samples. For example, the highest $\delta^{15}$N observed in the study was -2.7‰, and this



sample has a < 5% contribution from the mid-latitude zone. The other two low-latitude samples have 30% to 40% contribution
from the mid-latitude zone and their $\delta^{15}N$ is lower (Table S3), as expected due to the influence of $RONO_2$ emissions. The
influence of low $\delta^{15}N$-$NO_3^-$ from $RONO_2$ emissions is not limited to the Southern Ocean, and this estimate of the N isotopic
composition for the $RONO_2$ derived $NO_3^-$ source may be useful to constrain the contribution of $RONO_2$ emissions to $NO_3^-$
formation in other ocean regions with elevated surface ocean nitrite concentrations, such as the tropical Pacific.
**4.3) The O isotopes of atmospheric nitrate**
The corresponding $\delta^{18}O$ values allow us to determine the pathways of $NO_3^-$ formation from $NO_x$. However, an assumption
must first be made regarding the oxidation of NO to $NO_2$. While the dominant oxidant of NO to $NO_2$ is $O_3$ (R1) in most of the
troposphere, over the open ocean there can be a significant contribution via the reaction of NO with peroxy radicals ($HO_2$ and
its organic homologues $RO_2$) (Alexander et al., 2020). Peroxy radicals compete with $O_3$ to convert NO into $NO_2$ via R10:
$NO + HO_2$ (or $RO_2$) $\rightarrow NO_2 + OH$ (or RO)                                          (R10)
The $\delta^{18}O$ of peroxy radicals is much lower than that of $O_3$ because the O atoms derive from atmospheric $O_2$, which has a well-
defined $\delta^{18}O$ of 23.9‰ (Kroopnick & Craig, 1972). The $\delta^{18}O$-$NO_2$ can then be calculated using equation 2,
$\delta^{18}O\text{-}NO_2 = (\delta^{18}O\text{-}O_2)(1\text{-}f) + (\delta^{18}O\text{-}O_3{}^*)(f)$                                               (2)
where f is the fraction of $NO_2$ formed from R1, (1-f) is the fraction formed from R10, and the terminal $\delta^{18}O$-$O_3$ value ($\delta^{18}O$-
$O_3{}^*$) is 130.4 ± 12.9‰ (Vicars & Savarino, 2014).
The $\delta^{18}O$-$NO_3^-$ is then determined using equation 3 in which two thirds of the O atoms in $NO_3^-$ come from $NO_2$ and one
third comes from OH i.e., R3, or using equation 4 in which three sixths of the O atoms in $NO_3^-$ come from $O_3$, two sixths come
from $NO_2$ and one sixth comes from $H_2O$ i.e., R4-R6 (Hastings, et al., 2003; Alexander, et al., 2020).
$\delta^{18}O\text{-}NO_3^-{}_{(R3)} = (2/3)(\delta^{18}O\text{-}NO_2) + (1/3)(\delta^{18}O\text{-}OH)$                                  (3)
$\delta^{18}O\text{-}NO_3^-{}_{(R4\text{-}R6)} = (1/2)(\delta^{18}O\text{-}O_3{}^*) + (1/3)(\delta^{18}O\text{-}NO_2) + (1/6)(\delta^{18}O\text{-}H_2O)$                 (4)
We assume that 15% of NO to $NO_2$ conversion occurs via $HO_2$/$RO_2$ oxidation and 85% by $O_3$ oxidation as is suggested by
global models (Alexander, et al., 2020), and use the minimum and maximum $\delta^{18}O$-$H_2O$ range of -27.5‰ to 0‰, the
temperature-dependent equilibrium isotope exchange between OH and $H_2O$ (Walters & Michalski, 2016), and the resulting
minimum and maximum estimates for $\delta^{18}O$-OH of -67.4‰ to -41.0‰. Using these assumptions and equations 3 and 4, the
expected $\delta^{18}O$-$NO_3^-$ for the daytime OH oxidation pathway (R3) is 46.5‰ to 71.4‰, and for the dark reactions (R4-R6) is
88.7‰ to 113.5‰. The observed $\delta^{18}O$-$NO_3^-$ values were all less than 70‰ (Figs. 1C and 3), suggesting that $NO_x$ oxidation by
OH (R3) was indeed the dominant pathway for atmospheric $NO_3^-$ formation during summer. The low $\delta^{18}O$-$NO_3^-$ values
observed suggest a minimal influence of $O_3$ in the oxidation chemistry, ruling out both the halogen (R8 to R9) and DMS (R7)
related $NO_3^-$ formation pathways in addition to $N_2O_5$ hydrolysis (R4-6). This is consistent with previous year-round studies of
atmospheric $NO_3^-$ at coastal Antarctica (Savarino, et al., 2007) and the South Pole (Walters, et al., 2019) where $\delta^{18}O$-$NO_3^-$ was
at a minimum in summer (59.6‰ and 47.0‰, respectively). Both studies confirm the importance of $HO_x$ oxidation chemistry



in summer when solar radiation enhances the production of these oxidants, followed by a switch to $O_3$ dominated oxidation
chemistry in winter (Savarino, et al., 2007; Ishino, et al., 2017; Walters, et al., 2019).
Interestingly, most aerosol samples have a $\delta^{18}O\text{-}NO_3^-$ less than 46.5‰ (n=18), the lower limit estimated above for the OH
pathway. This suggests that there is more NO to $NO_2$ conversion via $HO_2/RO_2$ oxidation occurring than the global average. A
maximum $HO_2/RO_2$ contribution to NO oxidation of ~63% is required to explain the lowest $\delta^{18}O\text{-}NO_3^-$ value, which was
observed over the mid-latitudes during early summer. Increased $RO_2$ production over the mid-latitudes could derive from
$RONO_2$ photolysis in the MBL, which we hypothesis is happening in this region based on the $\delta^{15}N\text{-}NO_3^-$ (Section 4.2.2).
Although the lowest $\delta^{18}O$ observation occurred in the mid-latitudes, the majority of low $\delta^{18}O\text{-}NO_3^-$ values were observed in
the Weddell Sea, away from the region of maximum $RONO_2$ emissions. Approximately half of the Weddell Sea samples have
a $\delta^{18}O\text{-}NO_3^- < 31‰$, which would require a $HO_2/RO_2$ contribution to NO oxidation upwards of 40% (more than double the
contribution estimated by global models (Alexander, et al., 2020)). These $\delta^{18}O\text{-}NO_3^-$ observations are unusually low compared
to previous observations for the same region in spring (Morin, et al., 2009). We hypothesize that the large contribution of
$HO_2/RO_2$ to $NO/NO_2$ oxidation (i.e., a decrease in f in equation 2) resulting in these low $\delta^{18}O\text{-}NO_3^-$ values is due to the
influence of sea ice emissions. The 72-hour AMBTs for these low $\delta^{18}O\text{-}NO_3^-$ Weddell Sea samples indicate that all the air
masses either originated from, or spent a significant amount of time recirculating, over the sea ice covered region of the western
Weddell Sea (Fig. 6A). By contrast, aerosol samples from the Weddell Sea with $\delta^{18}O\text{-}NO_3^-$ values greater than 31‰ have air
masses that experienced significantly more oceanic influence (Fig. 6A). There is evidence that sea ice can lead to enhanced
peroxy radical production (Brough et al., 2019). In that work, increased $HO_2 + RO_2$ concentrations were observed during
spring at a coastal Antarctic site when air masses arrived from across a sea ice covered zone. This was attributed to the oxidation
of hydrocarbons by chlorine atoms, which leads to increased $RO_2$ concentrations via R11 and R12:
$RH + Cl \rightarrow R + HCL$                                                                                          (R11)
$R + O_2 \rightarrow RO_2$                                                                                          (R12)

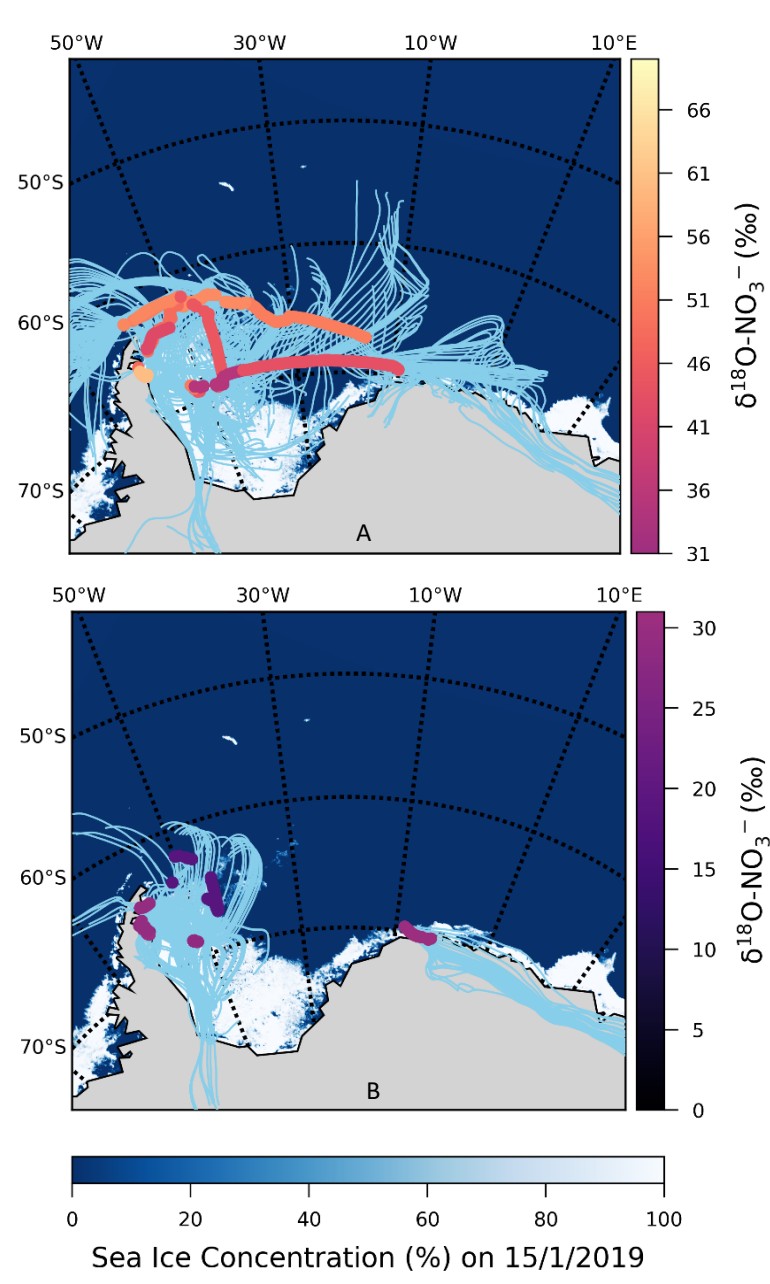

Fig. 6. 72-hour AMBT's (light blue dashed lines) computed for each hour of the voyage in the Weddell Sea, when the HV-AS was operational for more than 45 minutes of the hour. The vertical colour bar represents the weighted average $\delta^{18}O$ of atmospheric nitrate ($\delta^{18}O$-NO$_3^-$), where $\delta^{18}O$-NO$_3^-$ was > 31‰ (A) and < 31‰ (B). The horizontal colour bar represents satellite derived sea ice concentration (%) for the 15$^{th}$ of January 2019, which corresponds to midway through the WS sampling period. Satellite derived sea ice concentration was obtained from the AMSR2 ASI programme.



Cl atoms are much more reactive with hydrocarbons than OH (Monks, 2005) and can enhance hydrocarbon oxidation even
when present at low concentrations. Brough et al. (2019) suggest that air masses that traversed the sea ice zone contained
photolabile chlorine compounds that built up at night until photolysis occurred during the next day (Brough, et al., 2019).
Although our study was conducted in summer (the season of minimum sea ice extent), the sampling locations were uniquely
positioned at the western edge of the Weddell Sea gyre where significant sea ice remained (Fig. 6). Therefore, we suggest that
chlorine chemistry over the sea ice increased $RO_2$ concentrations at the time of our sampling, allowing the NO + $RO_2$ pathway
to play a more significant role in the Weddell Sea and resulting in low $\delta^{18}O$-$NO_3^-$ values. We note that the only other estimates
of $\delta^{18}O$-$NO_3^-$ from the Weddell Sea ranged from ~ 50‰ to 110‰ during springtime, and these samples were associated with
air masses that spent almost no time over the sea ice and therefore had limited potential for this peroxy radical chemistry to
drive down the $\delta^{18}O$-$NO_3^-$ to the low values we observed (Morin, et al., 2009).

**5) Conclusions**

Our observations across a large latitudinal gradient of the summertime Southern Ocean MBL suggest it is dominated by natural
$NO_x$ sources with unique isotopic signatures. Aerosol $NO_3^-$ was predominantly formed from lightning generated $NO_x$ with a
$\delta^{15}N$ of ~ 0‰ at the lower latitudes, whereas snowpack $NO_x$ emissions with a $\delta^{15}N$ ~ -48‰ dominated the MBL inventory at
higher latitudes. Over the mid-latitudes, $NO_3^-$ derived primarily from oceanic $RONO_2$ emissions, with an estimated $\delta^{15}N$
signature of ~ -22.0‰. This estimate may be valuable in constraining the contribution of oceanic $RONO_2$ emissions to $NO_3^-$
formation in other ocean regions where this source has been invoked, such as the tropical Pacific (Kamezaki et al., 2019). The
isotopic composition of $NO_3^-$ observed here can further inform interpretations of Antarctic ice core $NO_3^-$ isotope records to
understand aerosol climate forcing and controls on the atmospheric oxidation budget over millennia (Freyer, et al., 1996; Jiang,
et al., 2019) – the interpretation of which relies on knowledge of the $NO_x$ isotopic source signatures in the polar atmosphere.
The $\delta^{18}O$-$NO_3^-$ values were consistently lower than 70‰, which confirms $NO_x$ oxidation by OH (R3) to be the dominant
pathway for atmospheric $NO_3^-$ formation during summer. However, unusually low $\delta^{18}O$-$NO_3^-$ values observed at the mid-
latitudes and in the Weddell Sea indicate the increased importance of peroxy radicals (and decreased importance of $O_3$) in NO
oxidation to $NO_2$. At the mid-latitudes peroxy radicals ($RO_2$) may derive from  $RONO_2$ photolysis in the MBL, while In the
Weddell Sea, sea ice appears to play an important role in the formation of this oxidant via its influence on chlorine chemistry
in the MBL (Brough, et al., 2019). This implies that snow covered sea ice is not only a source of $NO_x$ but also other species
that have the potential to change the composition of the atmosphere above the ice and impact $NO_x$ oxidation chemistry. These
results also highlight the utility of $\delta^{18}O$-$NO_3^-$ to identify the major oxidants in NO oxidation, as well as $NO_x$ to $NO_3^-$ conversion.
In particular, $\delta^{18}O$-$NO_3^-$ can serve as a useful tool for testing our understanding of the relative importance of $HO_2$/$RO_2$ in
NO/$NO_2$ cycling, which can be difficult to constrain in some environments.





Our study challenges the traditional paradigm that considers the ocean as a passive recipient of N deposition, as the
Southern Ocean mid-latitude $NO_3^-$ source may derive almost entirely from oceanic $RONO_2$ emissions. In the tropical equatorial
Pacific atmosphere, Kamezaki et al. (2019) also suggested evidence for a low $\delta^{15}N$-$NO_3^-$ source derived from the ocean. In the
subtropical Atlantic Ocean MBL, Altieri et al. (2016) found that biogeochemical cycling in the surface ocean can directly
influence the lower atmosphere serving as a source of aerosol organic N and ammonium. This study suggests that the surface
waters of the Southern Ocean may also serve as a $NO_x$ source, ultimately resulting in $NO_3^-$ aerosol formation. As such, the
surface ocean may play a bigger role in atmospheric oxidative capacity over remote marine regions than previously thought.

*Author contributions.* K.E.A designed the study and sampling campaign, acquired funding and supervised the research. K.E.A
and J.G. provided financial and laboratory resources and assisted in data validation. K.A.M.S. and J.M.B conducted the
sampling at-sea and J.M.B. performed the laboratory analyses. M.G.H and E.J. assisted in data validation, reviewing, and
editing the manuscript. J.M.B analysed the data and prepared the manuscript with contributions from all co-authors.

*Competing interests.* The authors declare that they have no conflict of interest.

*Data availability statement.* Data sets for this research are available at 10.5281/zenodo.5006983

*Acknowledgements.*  We thank the Captain and crew of the R/V *SA Agulhas II* for their support at sea and the Marine
Biogeochemistry Lab in the Oceanography Department at the University of Cape Town for their assistance in the field and
laboratory. We thank Lija Treibergs, Reide Jacksin and Peter Ruffino for their assistance in analysing the nitrate isotopes and
Riesna Audh for her assistance with satellite derived sea ice concentration data. We thank Riesna Audh, Raquel Flynn and
Shantelle Smith for nitrite concentration measurements, and Raquel Flynn for quality controlling the nitrite concentration data.
This research was partially funded by a CAREER award to JG from the U.S. National Science Foundation (OCE-1554474).
This research was also supported by the South African National Research Foundation through a Competitive Support for Rated
Researchers Grant to K.E.A. (111716), South African National Antarctic Programme Postgraduate Fellowship to J.M.B, and
Grant to K.E.A (110732); UCT support to K.E.A. through a University Research Council Launching Grant and VC Future
Leaders 2030 Grant.
**6) References**
Alexander, B., & Mickley, L. J.: Paleo-perspectives on the potential future changes in the oxidative capacity of the atmosphere

due to climate change and anthropogenic emissions, Current Pollution Reports., 1, 57-69, DOI 10.1007/s40726-015-

0006-0, 2015.





Alexander, B., Sherwen, T., Holmes, C. D., Fisher, J. A., Chen, Q., Evans, A. J., Kasibhatla, P.: Global inorganic nitrate
production mechanisms: comparison of a global model with nitrate isotope observations. Atmospheric Chemistry and
Physics, 20(6), 3859-3877, https://doi.org/10.5194/acp-20-3859-2020, 2020.

Altieri, K. E., Fawcett, S. E., & Hastings, M.G.: Reactive Nitrogen Cycling in the Atmosphere and Ocean, Annual Review of
Earth and Plantetary Sciences, 49, 513-540, https://doi.org/10.1146/annurev-earth-083120-052147, 2021.

Altieri, K. E., Fawcett, S. E., Peters, A. J., Sigman, D. M., & Hastings, M. G.: Marine biogenic source of atmospheric organic
nitrogen in the subtropical North Atlantic, PNAS., 113(4), 925-930, https://doi.org/10.1073/pnas.1516847113, 2016

Altieri, K. E., Hastings, M. G., Gobel, A. R., Peters, A. J., & Sigman, D. M.: Isotopic composition of rainwater nitrate at
Bermuda: the influence of air mass source and chemistry in the marine boundary layer, Journal of Geophysical
Research: Atmospheres, 118, 11304-11316, https://doi.org/10.1002/jgrd.50829, 2013.

Atlas, E., Pollock, W., Greenberg, J., Heidt, L., & Thompson, A. M..: Alkyl nitrates, nonmethane hydrocarbons, and
halocarbon gases over the equatorial Pacific ocean during Saga 3, Journal of Geophysical Research Letters, 98(D9),
16933-16947, https://doi.org/10.1029/93JD01005, 1993.

Baker, A. R., Lesworth, T., Adams, C., Jickells, T. D., & Granzeveld, L.: Estimation of atmospheric nutrient inputs to the
Atlantic Ocean from 50°N to 50°S based on large-scale field sampling: Fixed nitrogen and dry deposition of
phosphorus, Global Biogeochemical Cycles, 24, GB3006, https://doi.org/10.1029/2009GB003634, 2010.

Bauguitte, A. J.-B., Bloss, W. J., Evans, M. J., Salmon, R. A., Anderson, P. S., Jones, A. E., Lee, J. D., Saiz-Lopez, A., Roscoe,
H. K., Wolff, E. W., &Plane, J. M. C.: Summertime NO$_x$ measurements during the CHABLIS campaign: can source
and sink estimates unravel observed diurnal cycles?, Atmospheric Chemistry and Physics, 12, 989-1002,
https://doi.org/10.5194/acp-12-989-2012, 2012.

Behrenfeld, M. J., Boss, E., Siegel, D. A., Shea, D. M.: Carbon-based ocean productivity and phytoplankton physiology from
space, Global Biogeochemical Cycles, 19(1), 1-14, https://doi.org/10.1029/2004GB002299, 2005.

Berhanu, T. A., Meusinger, C., Erbland, J., Jost, R., Bhattacharya, S. K., Johnson, M. S., & Savarino, J.: Laboratory study of
nitrate photolysis in Antarctic snow. II. Isotopic effects and wavelength dependence, The Journal of Chemical
Physics, 140(244306), 1-14, https://doi.org/10.1063/1.4882899, 2014.

Berhanu, T. A., Savarino, J., Erbland, J., Vicars, W. C., Preunkert, S., Martins, J. F., & Johnson, M. S.: Isotopic effects of
nitrate photochemistry in snow: a field study at Dome C, Antarctica, Atmospheric Chemistry and Physics, 15, 11243-
11256, https://doi.org/10.5194/acp-15-11243-2015, 2015.

Blake, N. J., Blake, D. R., & Swanson, A. L., Atlas, E., Flocke, F., & Rowland, F. S.: Latitudinal, vertical, and seasonal
variations of C1-C4 alkyl nitrate in the troposphere over the Pacific Ocean during PEM-Tropics A and B: Oceanic
and continental sources, Journal of Geophysical Research Letters, 108(D2), 1-14, doi:10.1029/2001JD001444, 2003.

Blake, N. J., Blake, D. R., Wingenter, O. W., Sive, B. C., Kang, C. H., Thornton, D. C., Bandy, A. R., Atlas, E., Flocke, F.,
Harris, J. M., & Rowland, F. S.: Aircraft measurements of the latitudinal, vertical, and seasonal variations of NMHCs,





methyl nitrate, methyl halides, and DMS during the First Aerosol Characterization Experiment (ACE 1), Journal of
Geophysical Research Letters, 104(D17), 21803-21817, 1999.
Bölhke, J. K., Mroczkowski, S. J., & Coplen, T. B.: Oxygen isotopes in nitrate: new reference materials for $^{18}$O:$^{17}$O:$^{16}$O
measurements and observations on nitrate-water equilibrium, Rapid Communications in Mass Spectrometry, 17(16),
1835-1846, https://doi.org/10.1002/rcm.1123, 2003.
Brough, N., Jones, A. E., & Griffiths, P. T.: Influence of sea ice-derived halogens on atmospheric $HO_x$ as observed in
Springtime coastal Antarctica, Geophysical Research Letters, 46, 10168-10176, https://doi.
org/10.1029/2019GL083825, 2019.
Casado, M., Landais, A., Masson-Delmotte, V., Genthon, C., Kerstel, E., Kassi, S., Arnaud, L., Picard, G., Prie, F., Cattani,
O., Steen-Larsen, H. -C., Vignon, E., & Cermak, P.: Continuous measurements of isotopic composition of water
vapour on the East Antarctic Plateau, Atmospheric Chemistry and Physics, 16(13), 8521-8538,
https://doi.org/10.5194/acp-16-8521-2016, 2016.
Casciotti, K.L., Sigman, D.M., Hastings, M.G., Böhlke, J.K. & Hilkert, A.: Measurement of the oxygen isotopic composition
of nitrate in seawater and freshwater using the denitrifier method, Analytical chemistry, 74(19), 4905-4912,
https://doi.org/10.1021/ac020113w, 2002.
Chuck, A. L., Turner, S. M., & Liss, P. S.: Direct evidence for a marine source of $C_1$ and $C_2$ alkyl nitrates, Science, 297, 1151-
1154, https://doi.org/10.1126/science.1073896, 2002.
Dahl, E. E., & Saltzman, S. E.: Alkyl nitrate photochemical production rates in North Pacific seawater, Marine Chemistry,
112, 137-141, https://doi.org/10.1016/j.marchem.2008.10.002, 2008.
Dahl, E. E., Heiss, E. M., & Murawski, K.: The effects of dissolved organic matter on alkyl nitrate production during GOMECC
and laboratory studies, Marine Chemistry, 142, 11-17, https://doi.org/10.1016/j.marchem.2012.08.001, 2012.
Dahl, E. E., Saltzman, E. S., & de Bruyn, W. J.: The aqueous phase yield of alkyl nitrates from ROO + NO: Implications for
photochemical production in seawater, Geophysical Research Letters, 30(6), 1-4,
https://doi.org/10.1029/2002GL016811, 2003.
Dahl, E. E., Yvon-Lewis, S. A., & S, S. E.: Saturation anomalies of alkyl nitrates in the tropical Pacific Ocean, Geophysical
Research Letters, 32(L20817), 1-4, https://doi.org/10.1029/2005GL023896, 2005.
Dar, S. S., Ghosh, P., Swaraj, A., & Kumar, A.: Graig-Gordon model validation using observed meteorological parameters
and measured stable isotope ratios in water vapor over the Southern Ocean, Atmospheric Chemistry and Physics
Discussions, 20(19), 11435-11449, https://doi.org/10.5194/acp-20-11435-2020, 2020.
Davidson, E. A., & Kingerlee, W.: A global inventory of nitric oxide emissions from soils, Nutrient Cycling in
Agroecosystems, 48, 37-50, 1997.
Elliot, E. M., Kendall, C., Wankel, S. D., Burns, S. A., Boyer, E. W., Harlin, K., Bain, D. J., & Butler, T. J.: Nitrogen isotopes
as indicators of $NO_x$ source contributions to atmospheric nitrate deposition across the Midwestern and Northeastern
United States . Environmental Science and Technology, 41(22), 7661-7667, https://doi.org/10.1021/es070898t, 2007.



Erbland, J., Vicars, W. C., Savarino, J., Morin, S., Frey, M. M., Frosini, D., Vince, E., Martins, J. M. F.: Air-snow transer of
nitrate on the East Antarctic Plateau - Part 1: Isotopic evidence for a phytolytically driven dynamic equilibrium in
summer. Atmospheric Chemistry and Physics, 13(13), 6403-6419, https://doi.org/10.5194/acp-13-6403-2013, 2013.

Fang, Y. T., Koba, K., Wang, X. M., Wen, D. Z., Li, J., Takebayashi, Y., Liu, X. Y., & Yoh, M.: Anthropogenic imprints on
nitrogen and oxygen isotopic composition of precipitation nitrate in a nitorgen-polluted city in southern China.
Atmospheric Chemistry and Physics, 11, 1313-1325, https://doi.org/10.5194/acp-11-1313-2011, 2011.

Finlayson-Pitts, B. J., & Pitts, J. N.: Chemistry of the upper and lower troposphere. San Diego, California: Academic Press,
2000.

Fisher, J. A., Atlas, E. L., Barletta, B., Meinardi, S., Blake, D. R., Thompson, C. R., Ryerson, T. B., Peischl, J., Tzompa-Sosa,
Z. A., & Murray, L. T.: Methyl, ethyl and propyl nitrates: global distribution and impacts on reactive nitrogen in
remote marine environments, Journal of Geophysical Research: Atmospheres, 123(21), 412-429,
https://doi.org/10.1029/2018JD029046, 2018.

Frey, M. M., Savarino, J., Morin, S., Erbland, J., & Martins, J. M.: Photolysis imprint in the nitrate stable isotope signal in
snow and atmosphere of East Antarctica and implications for reactive nitrogen cycling, Atmospheric Chemistry and
Physics, 9, 8681-8696, https://doi.org/10.5194/acp-9-8681-2009, 2009.

Freyer, H. D.: Seasonal variation of $^{15}N/^{14}N$ ratios in atmospheric nitrate species, Tellus B: Chemical and Physical Meterology
599            , 43(1), 30-44, 1991.

Freyer, H. D., Kley, D., Volz-Thomas, A., & Kobel, K.: On the interaction of isotopic exchange processes with photochemical
reactions in atmospheric oxides of nitrogen, Journal of Geophysical Research, 98(D8), 14791-14796, 1993.

Freyer, H. D., Kobel, K., Delmas, R. J., Kley, D., & Legrand, M. R.: First results of $^{15}N/^{14}N$ ratios in nitrate from alpine and
polar ice cores, Tellus B: Chemical and Physical Meterology, 48(1), 93-105,
https://doi.org/10.3402/tellusb.v48i1.15671, 1996.

Gobel, A. R., Altieri, K. E., Peters, A. J., Hastings, M. G., & Sigman, D. M.: Insights into anthropogenic nitrogen deposition
to the North Atlantic investigated using the isotopic composition of aersol and rainwater nitrate, Geophysical
Research Letters, 5977-5982, https://doi.org/10.1002/2013GL058167, 2013.

Grannas, A. M., Jones, A. E., Dibb, J., Ammann, M., Anastasio, C., Beine, H. J., Bergin, M., Bottenheim, J., Boxe, C. S.,
Carver, G., Chen, G., Crawford, J. H., Domine, F., Frey, M. M., Guzman, M. I., Heard, D. E., Hemig, D., Hoffmass,
610            M. R., Honrath, R. E., Huey, L. G., Hutterli, M., Jacobi, H. W., Klan, P., Lefer, B., McConnell, J., Plane, J., Sander,
R., Savarino, J., Shepson, P. B., Simpson, W. R., Sodeau, J. R., von Glasow, R., Weller, R., Wolff, E. W., & Zhu, T.:
An overview of snow photochemistry: evidence, mechanisms and impacts, Atmopsheric Chemistry and Physics
Discussions, 7(2), 4165-4283, https://doi.org/10.5194/acp-7-4329-2007, 2007.

Grasshoff, K., Kremling, K., & Ehrhardt, M.: Methods of seawater analysis, Verlag Chemi, Florida, 1983.





Guha, T., Lin, C. T., Bhattacharya, S. K., Mahajan, A. S., Ou-Yang, C.-F., Lan, Y.-P., Hsu, S. C., & Liang, M.-C.: Isotope
ratios of nitrate in aerosol samples from Mt. Lulin, a high altitude station in Central Taiwan, Atmospheric
Environment, 154, 53-69, http://dx.doi.org/10.1016/j.atmosenv.2017.01.036, 2017
Guilpart, E., Vimeux, F., Evan, S., Brioude, J., Mertzger, J., Barthe, C., Risi, C., & Cattani, O.: The isotopic composition of
near-surface water vaporat the Maido observatory (Reunion Island, southwestern Indian Ocean) documents the
controls of the humidity of the subtropical troposphere, Journal of Geophysical Research: Atmospheres, 122, 9628-
9650, 2017.

Hamilton, D. S., Lee, L. A., Pringle, K. J., Reddington, C. L., Spracklen, D. V., & Carslaw, K. S.: Occurence of pristine aerosol
environments on a polluted planet, Proceedings of the National Academy of Sciences, 111(52), 18466-18471,
https://doi.org/10.1073/pnas.1415440111, 2014.
Hastings, M. G., Sigman, D. M., & Lipschultz, F.: Isotopic evidence for source changes of nitrate in rain at Bermuda, Journal
of Geophysical Research: Atmospheres, 108(D24), https://doi.org/10.1029/2003JD003789, 2003.
Haywood, J., & Boucher, O.: Estimates of the direct and indirect radiative forcing due to tropospheric aerosols: a review,
Reviews of Geophysics, 513-543, 2000.
Hoering, T.: The isotopic composition of the ammonia and the nitrate ion in rain, Geochimica et Cosmochimica Acta, 12(1-
2), 97-102, 1957.

IPCC 2013: Boucher, O. D., Randall, P., Artaxo, C., Bretherton, G., Feingold, P., Forster, V.-M., Kerminen, Y., Kondo, H.,
Liao, U., Lohmann, P., Rasch, S.K., Satheesh, S., Sherwood, B., Stevens, & Zhang, X. Y.: Clouds and Aerosols, in:
Climate Change 2013: The Physical Science Basis. Contribution of Working Group I to the Fifth Assessment Report
of the Intergovernmental Panel on Climate Change, edited by: Stocker, T.F., Qin, D., Plattner, G.-K., Tignor, M.,
Allen, S. K., Boschung, J, Nauels, A., Xia, Y., Bex, V., & Midgley, P. M., Cambridge University Press, Cambridge,
United Kingdom and New York, NY, USA, 2013.
Ishino, S., Hattori, S., Savarino, J., Jourdain, B., Preunkert, S., Legrand, M., Caillon, N., Barbero, A., Kurlbayashi, N., &
Yoshida, N.: Seasonal variations of triple oxygen isotopic compositions of atmospheric sulfate, nitrate and ozone at
Durmont d'Urville, coastal Antarctica, Atmospheric Chemistry and Physics, 17, 3713-3727,
https://doi.org/10.5194/acp-17-3713-2017, 2017.
Jacobi, H.-W., & Schrems, O.: Peroxyacetyl nitrate (PAN) distribution over the South Atlantic Ocean, Physical Chemistry
Chemical Physics, 1, 5517-5521, https://doi.org/10.1039/A905290I, 1999.
Jacobi, H.-W., Weller, R., Jones, A. E., Anderson, P. S., & Schrems, O.: Peroxyaccetyl nitrate (PAN) concentrations in the
Antarctic troposphere measured during the photochemical experiment at Neumayer (PEAN'99). Atmospheric
Environment, 34, 5235-5247, https://doi.org/10.1016/S1352-2310(00)00190-4, 2000.
Jiang, S., Shi, G., Cole-Dai, J., Geng, L., Ferris, D. G., An, C., & Li, Y.: Nitrate preservation in snow at Dome A, East
Antarctica from ice core concentration and isotope records, Atmospheric Environment, 213, 405-412,
https://doi.org/10.1016/j.atmosenv.2019.06.031, 2019.



Johnston, J. C., & Thiemens, M. H. (1997). The isotopic composition of tropospheric ozone in three environments. Journal of
Geophysical Research: Atmospheres, 102(D21), 25395-25404.

Jones, A. E., Weller, R., Anderson, P. S., Jacobi, H.-W., Wolff, E. W., Schrems, O., & Miller, H.: Measurements of $NO_x$
emissions from the Antarctic snowpack, Geophysical Research Letters, 28(8), 1499-1502,
https://doi.org/10.1029/2000GL011956, 2001.

Jones, A. E., Weller, R., Minikin, A., Wolff, E. W., Sturges, W. T., McIntyre, H. P., Leonard, S. R., Schrems, O., & Bauguitte,
S.: Oxidised nitrogen chemistry and speciation in the Antarctic troposphere, Journal of Geophysical Research,
104(D17), 21355-21366, https://doi.org/10.1029/1999JD900362, 1999.

Jones, A. E., Weller, R., Wolff, E. W., & Jacobi, H.-W.: Speciation and rate of photochemical NO and $NO_2$ production in
Antarctic snow, Geophysical Research Letters, 27(3), 345-348, https://doi.org/10.1029/1999GL010885, 2000.

Kamezaki, K., Hattori, S., Iwamoto, Y., Ishino, S., Furutani, H., Miki, Y., Uematsu, M., Miura, K., & Yoshida, N.: Tracing
the sources and fromation pathways of atmospheric particulate nitrate over the Pacific Ocean using stable isotopes,
Atmospheric Environment , 209, 152-166, https://doi.org/10.1016/j.atmosenv.2019.04.026, 2019.

Kendall, C., Elliot, E. M., & Wankel, S. D.: Tracing anthropogenic inputs of nitrogen to ecosystems, in: Stable isotopes in
ecology and environmental science, edited by: Michener, R., & Lajtha, K., Blackwell Publishing, Malden, Mass 375-
449, https://doi.org/10.1002/9780470691854.ch12, 2007.

Kim, M. J., Michaud, J. M., Williams, R., Sherwood, B. P., Pomeroy, R., Azam, F., Burkart, M., & Bertram, T. H.: Bacteria-
driven production of alkyl nitrates in seawater, Geophysical Research Letters, 42, 597-604,
https://doi.org/10.1002/2014GL062865, 2015.

Krankowsky, D., Bartecki, F., Klees, G. G., Mauersberger, K., & Schellenbach, K.: Measurement of heavy isotope enrichment
in tropospheric ozone, Geophysical Research Letters, 22(13), 1713-1716, https://doi.org/10.1029/95GL01436, 1995.

Kroopnick, P., & Craig, H.: Atmospheric oxygen: isotopic composition and solubility fractionation, Science, 175(4017), 54-
55, 1972.

Lee, H-M., Henze, D. K., Alexander, B., & Murray, L. T.: Investigating the sensitivity of surface-level nitrate seasonality in
Antarctica to primary sources using a global model, Atmospheric Environment, 89, 757-767,
http://dx.doi.org/10.1016/j.atmosenv.2014.03.003, 2014.

Michalski, G., Bhattacharya, S. K., & Mase, D. F.: Oxygen isotope dynamics of atmospheric nitrate and its precursor molcules,
in: Handbook of environmental isotope geochemistry. Advances in Isotope Geochemistry, edited by: Baskaran, M.,
Springer, Berlin, Heidelberg, 613-635, https://doi.org/10.1007/978-3-642-10637-8_30, 2012.

Michalski, G., Scott, Z., Kabiling, M., & Thiemens, M. H.: First measurments and modeling of $\Delta^{17}O$ in atmospheric nitrate,
Geophysical Research Letters, 30(9), https://doi.org/10.1029/2003GL017015, 2003.

Monks, P. S.: Gas-phase radical chemistry in the troposphere, Chemical Society Reviews, 34, 376-395, DOI:
10.1039/b307982c, 2005.



Morin, S., Savarino, J., Frey, M. M., Domine, F., Jacobi, H. W., Kaleschke, L., & Martins, J. M.: Comprehensive isotopic
composition of atmospheric nitrate in the Atlantic Ocean boundary layer from 65°S to 79°N, Journal of Geophysical
Research, 114(D05303), 1-19, https://doi.org/10.1029/2008JD010696, 2009.

Nadzir, M. S., Ashfold, M. J., Khan, M. F., Robinson, A. D., Bolas, C., Latif, M. T., Wallis, B. M., Mead, M. I., Hamid, H. H.
686          A., Harris, N. R. P., Ramly, Z. T. A., Lai, G. T., Liew, J. N., Ahamed, F., Uning, R., Samah, A. A., Maulud, K. N.,
Suparta, W., Zainudin, S. K., Wahab, M. I. A., Sahani, M., Muller, M., Yeok, F. S., Rahman, N. A., Mujahid, A.,
Morris, K. I. & Sasso, N. D.: Spatial-temporal variations in surface ozone over Ushuaia and the Antarctic region:
observations from in situ measurements, satellite data, and global models, Environmental Science and Pollution
Research, 25, 2194-2210, https://doi.org/10.1007/s11356-017-0521-1, 2018.

Nesbitt, S. W., Zhang, R., & Orville, R. E.:Seasonal and global NOx production by lightning estimated from the Optical
Transient Detector (OTD), Tellus B: Chemical and Physical Meteorology, 52(5), 1206-1215,
https://doi.org/10.3402/tellusb.v52i5.17098, 2000.

Park, S. S., & Kim, Y. J.: Source contributions to fine particulate matter in an urban atmosphere, Chemosphere, 59, 217-226,
https://doi.org/10.1016/j.chemosphere.2004.11.001, 2005.

Park, Y., Park, K., Kim, H., Yu, S., Noh, S., Kim, M.-S, Kim, J.-Y., Ahn, J.-Y., Seok, K.-S., & Kim, Y.-H.: Characterizing
isotopic compositions of TC-C, $NO_3^-$-N and $NH_4^+$-N in $PM_{2.5}$ in South Korea: Impact of China's winter heating,
Environmental Pollution, 233, 735-744, https://doi.org/10.1016/j.envpol.2017.10.072, 2018.

Rolph, G.D.: Real-time Environmental Applications and Display System (READY) Website (http://www.ready.noaa.gov).
NOAA Air Resources Laboratory, College Park, MD, 2016.

Savarino, J., Kaiser, J., Morin, S., Sigman, D. M., & Thiemens, M. H.: Nitrogen and oxygen isotopic constraints on the origin
of atmospheric nitrate in coastal Antarctica, Atmopsheric Chemistry and Physics, 7, 1925-1945,
https://doi.org/10.5194/acp-7-1925-2007, 2007.

Schumann, U., & Huntrieser, H.: The global lightning induced nitrogen oxides source, Atmospheric Chemistry and Physics
Discussions, 7(1), 2623-2818, https://doi.org/10.5194/acp-7-3823-2007, 2007.

Shi, G., Buffen, A. M., Hastings, M. G., Li, C., Ma, H., Li, Y., Sun, B., An, C., & Jiang, S.: Investigating the post-depositional
processing of nitrate in East Antarctic snow: isotopic contraints in photolytic loss, re-oxidation, and source inputs,
Atmospheric Chemistry and Physics, 15, 9435-9453, https://doi.org/10.5194/acp-15-9435-2015, 2015.

Shi, G., Buffen, A. M., Ma, H., Hu, Z., Sun, B., Li, C., Yu, J., Ma, T., An, C., Jiang, S., Li, Y., & Hastings, M. G.:
Distinguishing summertime atmopsheric production of nitrate across the East Antarctic ice sheet, Geochimica et
Cosmochimica Acta , 231, 1-14, https://doi.org/10.1016/j.gca.2018.03.025, 2018.

Shi, G., Ma, H., Zhu, Z., Hu, A., Chen, Z., Jiang, Su., An, C., Yu, J., Ma, T., Li, Y., Sun, B., Hastings, M. G.: Using stable
isotopes to distinguish atmospheric nitrate production and its contribution to the surface ocean across hemispheres,
Earth and Planetary Science Letters, 564, 1-12, https://doi.org/10.1016/j.epsl.2021.116914, 2021.



Sigman, D.M., Casciotti, K.L., Andreani, M., Barford, C., Galanter, M.B.J.K. & Böhlke, J.K.: A bacterial method for the
nitrogen isotopic analysis of nitrate in seawater and freshwater, Analytical chemistry, 73(17), 4145-4153,
https://doi.org/10.1021/ac010088e, 2001.
Spreen, G., Kaleschke, L., & Heygster, G.: Sea ice remote sensing using AMSR-E 89-GHz channels, Journal of Geophysical
Research: Oceans, 113(C02S03), 1-14, https://doi.org/10.1029/2005JC003384, 2008.
Stein, A.F., Draxler, R.R, Rolph, G.D., Stunder, B.J.B., Cohen, M.D., & Ngan, F.: NOAA's HYSPLIT atmospheric transport
and dispersion modeling system, Bull. Amer. Meteor. Soc., 96, 2059-2077, https://doi.org/10.1175/BAMS-D-14-
00110.1, 2015.

Thiemens, M. H.: History and applications of mass-independent isotope effects. Annual Review of Earth and Planetary
Sciences, 34, 217-262, https://doi.org/10.1146/annurev.earth.34.031405.125026, 2006.
van der A, R. J., Eskes, H. J., Boersma, K. F., van Noije, T. P., Van Roozendael, M., De Smedt, I., Peters, D. H. M. U., &
Meijer, E. W.: Trends, seasonal variability and dominant $NO_x$ source derived from a ten year record of $NO_2$ measured
from space, Journal of Geophysical Research, 113, 1-12, https://doi.org/10.1029/2007JD009021, 2008.
Vicars, W. C., Savarino, J.: Quantitative constraints on the $^{17}O$-excess ($\Delta^{17}O$) signature of surface ozone: Ambient
measurements from 50°N to 50°S using the nitrite-coated filter technique. Geochimica et Cosmochimica Acta, 135,
270-287, https://doi.org/10.1016/j.gca.2014.03.023, 2014.
Virkkula, A., Teinila, K., Hillamo, R., Kerminen, V. M., Saarikoski, S., Aurela, M., Viidanoja, J., Paatero, J., Koponen, I. K.,
& Kulmala, M.: Chemical composition of boundary layer aerosol over the Atlantic Ocean and at an Antarctic site,
Atmospheric Chemistry and Physics, 6(11), 3407-3421, https://doi.org/10.5194/acp-6-3407-2006, 2006.
Walters, W. W., & Michalski, G.: Theoretical calculation of nitorgen isotope equilibrium exchange fractionation factors for
various $NO_y$ molecules, Geochimica et Cosmochimica Acta, 164, 284-297,
http://dx.doi.org/10.1016/j.gca.2015.05.029, 2015.
Walters, W. W., & Michalski, G.: Theoretical calculation of oxygen equilibrium isotope fractionation factors involving various
$NO_y$ molecules, OH, and $H_2O$ and its implications for isotope variations in atmospheric nitrate, Geochimica et
Cosmochimica Acta, 191, 89-101, http://dx.doi.org/10.1016/j.gca.2016.06.039, 2016.
Walters, W. W., Michalski, G., Bohlke, J. K., Alexander, B., Savarino, J., & Thiemens, M. H.: Assessing the seasonal dynamics
of nitrate and sulfate aerosols at the South Pole utilizing stable isotopes, Journal of Geophysical Research:
Atmospheres, 124(14), 8161-8177, https://doi.org/10.1029/2019JD030517, 2019.
Walters, W. W., Simonini, D. S., & Michalski, G.: Nitrogen isotope exchange between NO and $NO_2$ and its implications for
$\delta^{15}N$ variations in tropospheric $NO_x$ and atmospheric nitrate, Geophysical Research Letters, 43, 440-448,
https://doi.org/10.1002/2015GL066438, 2016.
Weller, R., Jones, A. E., Wille, A., Jacobi, H.-W., McIntyre, H. P., Sturges, W. T., Huke, M., & Wagenback, D.: Seasonality
of reactive nitrogen oxides ($NO_y$) at Neumayer Station, Antarctica, Journal of Geophysical Research, 107(D23), 1-
11, https://doi.org/10.1029/2002JD002495, 2002.



Williams, J. E., Le Bras, G., Kukui, A., Ziereis, H., Brenninkmeijer, C. A. M.: The impact of the chemical production of methyl
nitrate from the NO + $CH_3O_2$ reaction on the global distributions of alkyl nitrates, nitrogen oxides and tropospheric
ozone: a global modelling study. Atmospheric Chemistry and Physics, 14(5), 2363-2382, https://doi.org/10.5194/acp-
14-2363-2014, 2014.

Yeatman, S. G., Spokes, P. F., Dennis, P. F., & Jickells, T. D.: Camparisons of aerosol nitrogen isotopic composition at two
polluted coastal sites, Atmospheric Environments, 35, 1307-1320, https://doi.org/10.1016/S1352-2310(00)00408-8,
2001.

Zong, Z., Wang, X., Tian, C., Chen, Y., Fang, Y., Zhang, F., Li, C., Sun, J., Li, J., & Zhang, G.: First assessment of NOx
sources at a regional background site in North China using isotopic analysis linked with modeling, Environmental
Science and Technology, 51, 5923-5931, https://doi.org/10.1021/acs.est.6b06316, 2017.
