# Peer review of "The importance of alkyl nitrates and sea ice emissions to atmospheric NOx sources and cycling in the summertime Southern Ocean marine"

_Atmospheric Chemistry and Physics, 2021_

## Referee Comment (RC2)

Review of DOI.org/10.5194/acp-2021-519

General Comments

In this paper the authors present a very interesting and unique set of data on the isotopic composition of coarse mode atmospheric nitrate in the Southern Ocean marine boundary layer.  The data were derived from latitudinal transects as well as sampling near the Weddell Sea and Ronne-Filchner Ice Shelf. The authors combined their isotopic measurements of aerosol nitrate with air-mass back trajectory (AMBT) analysis to determine the major sources of NOx contributing to nitrate aerosols in the Southern Ocean. The paper is well written, the data are clearly presented, and the conclusions derived from the study are well supported, with the possible exception of the estimated $\delta^{15}N$ signature of oceanic nitrate emissions (see below).  I discuss a couple of potential technical issues with the isotope measurements below.  I also think that the authors can do more with their dataset – it seems a little over aggregated. For example, I am curious to know how the relationship between $\delta^{18}O$ and $\delta^{15}N$ varies in the study and if the different size fractions of aerosol had different N masses and isotopic values.

Specific Comments:

Lines 88-131. This is a very nice introduction to the main reactions affecting the isotopic composition of oxidized N in the atmosphere.

Line 150-151.  State the specific Tisch Environmental filter used along with its surface area.

Line 171. Ultra-clean deionized water is vague.  Please state the resistance of the DI water in megaohms.

Lines 178-179. I am confused by the presentation of the filter blanks.  The text states that the average amount of nitrate on a field-blank filter was 484.7 nmoles or 0.4847 µmoles (also, this value should have a standard deviation accompany it).  Since the filters were extracted in 25 ml of DI water, 0.4847 µmoles of nitrate would result in a nitrate concentration of 19.4 µmoles/liter which seems very high for a blank. I suspect that only a fraction of the blank filter was extracted in DI (hence the approximately 30 cm$^2$ reference in line 172) and if this is the case it should be included in the methods section to avoid confusion.  Later the authors state that "The pooled standard deviation (Sp) of four repeated sample measurements for [NO$_3^-$] was 0.3 µmol L$^{-1}$". Does this sentence refer to an actual aerosol sample or are the authors referring to the blank filters? If the sentence refers to blank filters, then it seems that only four blanks were run over several weeks of aerosol sampling. For context, the authors should state the range of nitrate values measured in the actual aerosol sample extracts.

Lines 188-190.  I believe it is more common for investigators to use the USGS 32 isotope standard (+180 per mil) and USGS 34 (-1.8 per mil) for calibration of $\delta^{15}N$ values measured with the microbial denitrifier method. In the present study IAEA N3 (+4.7) was substituted for USGS 32 resulting in a very truncated isotopic baseline for calibration.  I wonder if the authors could explain why they chose N3 over USGS 32 and if they think their $\delta^{15}N$ values are comparable to those in other papers.

Lines 195-197.  For each filter deployment, the authors weighted the isotopic values of the size fractions based on the mass of nitrate in the fractions.  This seems very reasonable and it simplifies the data

presentation. However, I am curious if there were any consistent mass or isotopic patterns in the size fractions. I believe a small table and a few lines of text would be quite valuable to readers.

Lines 198-200. Why were only nitrite concentrations measured in seawater (what about nitrate)? Also, it seems like isotopic measurements of ocean water $NO_2$ and or $NO_3$ would be useful for the later discussion surrounding $RONO_2$ as a source of atmospheric N.

Lines 217-219. State the latitudinal range of values used in the comparison of Weddell Sea and transect samples.

Line 229/Figure 1. Figure 1 is a good overview of the data, but I was curious about how these relationships varied when the data were disaggregated. Fortunately the authors provided a very nice table of data in the Supplemental Materials. Here are some additional graphs that I made with the data that bolster several of the authors hypotheses in the Discussion.

[Figure]

This figure clearly shows a strong relationship exists between nitrate concentration and $\delta^{15}N$ with the highest concentrations found in samples with the $\delta^{15}N$ near zero per mil. This finding supports the hypothesis that NOx produced by lightning is an important source of $NO_3$ aerosol in the Southern Ocean, especially in the latitude 30-40 degree band.

Using this plot, one can delineate samples without any clear influence from lightning ($\delta^{15}N < -12$ per mil). Looking at only samples without lightning influence, I then plotted dual-isotope graphs ($\delta^{15}N$ vs. $\delta^{18}O$) for the Weddell Sea and Transect datasets to see if there were any relationships detectable. Typically, dual isotope plots can be used to infer two endmember mixing dynamics in systems where fractionation is not important.

[Figure]

There is a significant linear relationship between $\delta^{15}N$ and $\delta^{18}O$ in the Weddell Seas samples, strongly suggesting two endmember mixing dynamics are at play. One could assign one endmember with $\delta^{15}N$ of ~+15 per mil and $\delta^{18}O$ of ~+25 per mil which corresponds to the hypothesized $RONO_2$ endmember discussed in the paper. A second endmember with relatively high $\delta^{18}O$ (~+70 per mil) and relatively low $\delta^{15}N$ (~ -40 per mil) can be defined and corresponds to the snow/ice source discussed in the paper.

A linear relationship was also observed between δ15N and δ18O for Transect samples.

[Figure]

Lines 243, 285 and elsewhere. Try to avoid the use of "very". When stating something is high or low it is probably better to say "relatively high" or "relatively low" based on your ranges of data values.

Line 250. It might be useful for readers if you gave a range of $NO_3$ concentrations in urban airsheds for purpose of comparison.

Line 255. How is lifetime defined?

Lines 265-266. Keeping the units the same might make this comparison clearer.

Lines 260-274.  The arguments against equilibrium fractionation having a role in the isotopic dynamics of atmospheric NOx are logical and convincing.

Line 281. Delete word "therefore".

Line 286. I suggest changing "lightning" to "thunderstorms" and "this mechanism" to "lightning".

Line 299. Remove word "previously".

Line 317.  Suggest changing "unlikely to be influenced" to "likely to be less influenced".

Section 4.2.2. Based solely on the stable isotope data presented, there does not need to be more than two endmembers to explain the range of $\delta^{15}N$ values observed in the study: i) near zero per mil lightning source and ii) relatively low, -48 to -56 per mil $NO_3$ produced from photolysis from snow/ice.  However, the authors have combined the isotope data with AMBT to gain better understanding of the atmospheric dynamics during their cruise. This analysis suggests that ocean-derived NOx, with intermediate $\delta^{15}N$ (~-22 per mil), could be an important atmospheric source in the mid-latitude samples. They then hypothesize that $RONO_2$ emissions are a major source of NOx in the Southern Ocean MBL.

I think this hypothesis needs some beefing up. First, the authors should discuss the accuracy of the AMBTs – are the models reliable in the Southern Ocean? Second, I think the authors should describe the biotic and abiotic processes that generate oceanic $RONO_2$ and the substrates on which these reactions rely. Third, it would be very useful to discuss both the range of $\delta^{15}N$ of the substrates ($NO_2$ and $NO_3$?) as well as the direction and magnitude of the N fractionations resulting from the production of oceanic $RONO_2$. While the authors did not measure $\delta^{15}N$ of $NO_2$ and $NO_3$ in their study, there are likely other data available from the Southern Ocean to include. Basically, what I'd like to see is if it is reasonable for $RONO_2$ to have a $\delta^{15}N$ of -22 +/- 7.5 per mil based on what we know about the substrates and reactions that produce it.

Line 467. I suggest replacing "unique" with distinct.

---

## Author Comment (AC1)

Response to Reviewer comments

We thank the reviewer for their supportive and constructive comments on the manuscript. We feel that the paper has been improved by the review process. Below, we address each of the reviewer's specific and technical line-by-line comments. The reviewer comments are in black text while the responses are in *blue italics.*

General Comments

In this paper, the authors present stable nitrogen and oxygen isotope measurements of aerosol nitrate collected on filter samples during 2018 and 2019 in the Southern Ocean between South Africa and Antarctica, and in the Weddell Sea. The measurements are then analyzed based on previous literature-based fractionation and isotope information on both nitrate nitrogen and oxygen sources in the different key study regions, including anthropogenic NOx, lightning NOx, emissions of RONO2, and snow-NOx emission, as well as O3 and other oxygen sources and equilibrium impacts. The introduction is thorough and well-written, and presents a clear context for the findings described in following sections. I find this paper and its findings compelling, and certainly worthy of publication, but there are a number of issues that need to be addressed prior to publication.

As well, there are a fair number of typographical errors and style guide recommendations that need to be addressed, and I would encourage the authors to take greater care in the future to address these details prior to manuscript submission, in particular for journals where there is no typesetting prior to the review process.

**Specific Comments**

Line 95 – Technically, $É\square = (KIE − 1) × 1000‰$, where KIE, the kinetic isotope effect is the ratio of the rates, $= k_{(primary N isotopologue)}/k_{(stable 15N isotopologue)}$, so $É›$ is the ratio – $1 × 1000‰$.

*This technicality will be amended in the text. The sentence on line 95 will read "In contrast, snow emitted $NO_x$ typically has a very low $\delta^{15}N$ signature due to the large fractionation ($^{15}\varepsilon$) of $\sim$ - 48‰ (Berhanu et al., 2014 and 2015) associated with $NO_3^-$ photolysis in the snowpack, where $^{15}\varepsilon = (KIE - 1) x 1000‰$ and the kinetic isotope effect (KIE) is the ratio of the rates with which the two isotopes of N are converted from reactant to product."*

Line 142 – the date range could be simplified by writing 7-19 December 2018. Similarly, on Line 143, you could write 4 January to 21 February 2019. Also, at some point, and this is probably the ideal time, it should be stated what time zone (UTC? local time? South Africa Standard Time?) is being used to describe the dates for the legs and the times for the filter sample collections.

*Both date ranges will be edited in the text. The time zone used to describe the dates for the cruise legs and the times for the filter sample collections was GMT. This will be included in the text as recommended by the reviewer.*

Lines 161-162 – Were the samples that were taken for less than 24 hours due to stagnant conditions due to unusual ship manoeuvres? Or was it a combination of both stagnant conditions and unusual ship manoevres? Also, Table S1 lists "Daylight (hrs)" (which should be "hours" or "h") but not total sampling duration times. So the reader isn't left doing their own calculations based on the location and time of year relative to the equinox, perhaps the authors could include both "total daylight hours sampled" and "total sampling duration" in Table S1. It would also be good to include sampling start and end times (with a reference time zone) in the table, rather than just the start day and stop day, to demonstrate consistency with the 13-88 hours reported in Line 162.

*Samples that were taken for less than 24 hours were done so if either the ship was stationary, resulting in stagnant conditions, or if unusual ship manoeuvres occurred, as both could have potentially led to contamination of our samples. This will be clarified in the text.*

*Table S1 will be amended to include the sampling duration in hours (h) to demonstrate consistency with the 13 to 88 hours reported in Line 162. To clarify, the daylight hours are the number of daylight hours that were available during the time of sampling, not the number of daylight hours that we explicitly sampled. We do not refer to this data in the main text and have therefore decided to remove this column for simplicity and to avoid confusion.*

*The high-volume air sampler was controlled by a sector collector, therefore the pump would only switch on when the wind was blowing from the front of the ship for a certain amount of time at a certain speed, as outlined in the methods. This will be reiterated in section 4.1 to remind the reader. As a result, the high-volume air sampler was not operating for the entire duration of every filter deployment and the sampling duration is not equivalent to subtracting the end times from the start times. Instead, the sampling duration was determined by the difference in the pump's hour meter reading before and after a filter sample deployment. A footnote will be added to supplementary Table 1 to clarify this.*
*Considering this and the fact the many of the filter deployments span multiple days, we do not think that it is necessary to include the time at which each sample was deployed and collected.*

Line 168 – for consistency with line 164 "field blank", you could write "field blank filters" or "field blank set of filters".

*This will be edited to read "field blank filters" as recommended.*

Lines 175-183 – I have questions about the filters. What is the precision of the IC system analysis? Is it 0.3 μmol/L? What was the reasoning behind the other analysis, and why was this only done on a subset of the filters? In line 183, the authors suggest the average [NO3-] is "reported", but where? Maybe point to it here? And why not report both? And are they labeled as this or that or both? There should be a notation in Table S3 for the samples that were analyzed using both methods.

*The precision of the IC system analysis is 0.3 μmol L⁻¹. This will be clarified in the text.*

*Our initial plan was to measure all of our samples using ion chromatography (IC) to determine nitrate concentration as well as other inorganic ion concentrations that are outside the scope of this manuscript. However, at the time of our sample analyses the IC was not operational. To prevent any*

*delays in analysis of the samples for isotopic composition, we decided to use the Lachat QuikChem®
flow injection autoanalyzer to determine the nitrate concentrations. This technique does not however
provide information on any of the other potentially important inorganic ions (i.e., ammonium, sulfate,
sodium, chloride, etc.), which are not reported here but will be used in other studies.*

*Once the IC system was operational, we performed a second separate series of filter extractions using
the left-over filter samples to measure the other cations and anions inclusive of nitrate. We thus
thought it most accurate to report an average of the two measurement techniques. We did not report
both nitrate concentrations measured using each technique separately, as we did not see a methods
comparison to be pertinent to this manuscript, especially considering that the average value was used
in all further data analysis.*

*Furthermore, separate filter extractions were used for each measurement technique, such that any
difference in concentration is likely driven by heterogeneity on the filter as opposed to the technique
used to measure the extract concentration.*

*Our methods inaccurately state that a subset of aerosol samples was analysed for nitrate
concentration using the Lachat QuikChem® flow injection autoanalyzer, when in fact all samples
were measured using both methods. The methods section will be amended to reflect this correctly. We
will also refer to the table when the nitrate concentration is reported in the text.*

*Furthermore, a conversion error was found in the determination of the volume of air sampled during
each filter deployment. This only effects the atmospheric $NO_3^-$ concentration (ng m$^{-3}$) and not the
isotope data. The atmospheric $NO_3^-$ concentration is reduced compared to the original manuscript;
however, the same trends are observed with latitude. We have updated the necessary figures, tables,
and online data repository.*

Line 198 – Where are the seawater samples provided? If you're not providing a table or
documentation on the seawater sample data, it would be prudent to provide a link at this
point, and not just at the end of the manuscript. Also, what depth does "position at depth
± 5 m" imply? Was the depth location actually uncertain to within a 10 m range? Lastly,
the reported data repository on line 500 is an incomplete link that goes nowhere, so this
needs to be sorted out prior to publication, or a final review of the paper.

*The data link has been updated: https://doi.org/10.5281/zenodo.5006983*
*We will edit the text to reflect the depth of ships underway system as "approximately 5 m", because
the depth at which the underway inlet sits is not actually uncertain to within a 10 m range as queried
by the reviewer.*

*We will also include a supplementary table that highlights the seawater data as shown below.*

*Table SX: The average (Avg) and standard deviation (SD) sea surface nitrite concentration ([$NO_2^-$]
(µmol L$^{-1}$)) measured during the early summer (ES) and late summer (LS) cruise transects. The date
(dd/mm/yyyy), time (GMT), latitude ($^o$ S) and longitude ($^o$ E) of each sample is also given.*

| Leg | Date | Time | Latitude | Longitude | [$NO_2^-$] (µmol L$^{-1}$) | |
|-----|------|------|----------|-----------|---------|-----|
| | (dd/mm/yyyy) | (GMT) | (° S) | (° E) | Avg | SD |
| ES | 07/12/2018 | 10:00:00 | 34.23 | 17.85 | 0.13 | 0.03 |

| ES | 07/12/2018 | 14:00:00 | 34.50 | 17.09 | 0.11 | 0.01 |
|----|------------|----------|-------|-------|------|------|
| ES | 07/12/2018 | 18:00:00 | 34.50 | 16.17 | 0.07 | 0.01 |
| ES | 07/12/2018 | 22:00:00 | 34.50 | 15.19 | 0.09 | 0.03 |
| ES | 08/12/2018 | 02:00:00 | 34.77 | 14.44 | 0.15 | 0.00 |
| ES | 08/12/2018 | 06:00:00 | 35.43 | 13.93 | 0.07 | 0.00 |
| ES | 08/12/2018 | 10:00:00 | 36.06 | 13.44 | 0.13 | 0.05 |
| ES | 08/12/2018 | 14:00:00 | 36.64 | 12.99 | 0.11 | 0.04 |
| ES | 08/12/2018 | 18:00:00 | 37.22 | 12.54 | 0.21 | 0.09 |
| ES | 08/12/2018 | 22:00:00 | 37.84 | 12.04 | 0.16 | 0.00 |
| ES | 09/12/2018 | 02:00:00 | 38.57 | 11.46 | 0.10 | 0.02 |
| ES | 09/12/2018 | 06:00:00 | 39.30 | 10.88 | 0.15 | 0.02 |
| ES | 09/12/2018 | 10:00:00 | 39.98 | 10.32 | 0.21 | 0.01 |
| ES | 09/12/2018 | 14:00:00 | 40.70 | 9.73 | 0.20 | 0.02 |
| ES | 09/12/2018 | 18:00:00 | 41.41 | 9.14 | 0.16 | 0.01 |
| ES | 09/12/2018 | 22:00:00 | 42.10 | 8.56 | 0.14 | 0.00 |
| ES | 10/12/2018 | 02:00:00 | 42.82 | 7.95 | 0.20 | 0.02 |
| ES | 10/12/2018 | 06:00:00 | 43.00 | 7.79 | 0.15 | 0.01 |
| ES | 10/12/2018 | 10:00:00 | 43.31 | 7.53 | 0.30 | 0.12 |
| ES | 10/12/2018 | 14:00:00 | 44.00 | 6.92 | 0.29 | 0.03 |
| ES | 10/12/2018 | 18:00:00 | 44.73 | 6.29 | 0.39 | 0.04 |
| ES | 10/12/2018 | 22:00:00 | 45.30 | 5.78 | 0.37 | 0.03 |
| ES | 11/12/2018 | 02:00:00 | 45.80 | 5.33 | 0.30 | 0.00 |
| ES | 11/12/2018 | 06:00:00 | 46.29 | 4.90 | 0.31 | 0.01 |
| ES | 11/12/2018 | 10:00:00 | 46.77 | 4.45 | 0.33 | 0.03 |
| ES | 11/12/2018 | 14:00:00 | 47.32 | 3.94 | 0.32 | 0.02 |
| ES | 11/12/2018 | 18:00:00 | 47.86 | 3.44 | 0.30 | 0.01 |
| ES | 11/12/2018 | 22:00:00 | 48.35 | 2.99 | 0.31 | 0.02 |
| ES | 12/12/2018 | 02:00:00 | 48.82 | 2.54 | 0.31 | 0.02 |
| ES | 12/12/2018 | 06:00:00 | 49.24 | 2.14 | 0.25 | 0.00 |
| ES | 12/12/2018 | 10:00:00 | 49.69 | 1.71 | 0.31 | 0.01 |
| ES | 12/12/2018 | 14:00:00 | 50.09 | 1.31 | 0.29 | 0.05 |
| ES | 12/12/2018 | 18:00:00 | 50.60 | 0.82 | 0.24 | 0.01 |
| ES | 12/12/2018 | 22:00:00 | 51.12 | 0.30 | 0.33 | 0.03 |
| ES | 13/12/2018 | 02:00:00 | 51.73 | 0.00 | 0.23 | 0.04 |
| ES | 13/12/2018 | 06:00:00 | 52.50 | 0.00 | 0.22 | 0.02 |
| ES | 13/12/2018 | 10:00:00 | 53.30 | 0.00 | 0.24 | 0.03 |
| ES | 13/12/2018 | 14:00:00 | 54.00 | 0.00 | 0.32 | 0.01 |
| ES | 13/12/2018 | 18:00:00 | 54.48 | 0.00 | 0.31 | 0.01 |
| ES | 13/12/2018 | 22:00:00 | 55.28 | -0.06 | 0.25 | 0.00 |
| ES | 14/12/2018 | 02:00:00 | 56.06 | 0.00 | 0.21 | 0.02 |
| ES | 14/12/2018 | 06:00:00 | 56.89 | 0.00 | 0.30 | 0.01 |
| ES | 14/12/2018 | 10:00:00 | 57.70 | -0.01 | 0.32 | 0.02 |
| ES | 14/12/2018 | 14:00:00 | 58.52 | 0.00 | 0.30 | 0.00 |
| ES | 14/12/2018 | 18:00:00 | 59.35 | -0.01 | 0.29 | 0.00 |
| ES | 14/12/2018 | 22:00:00 | 59.83 | -0.01 | 0.34 | 0.00 |
| ES | 15/12/2018 | 02:00:00 | 60.38 | -0.05 | 0.32 | 0.01 |

| | | | | | | |
|-----|-------------|----------|--------|--------|------|------|
| ES | 15/12/2018 | 06:00:00 | 61.10 | 0.00 | 0.28 | 0.00 |
| ES | 15/12/2018 | 10:00:00 | 61.58 | -0.01 | 0.29 | 0.04 |
| ES | 15/12/2018 | 14:00:00 | 62.30 | 0.00 | 0.28 | 0.02 |
| ES | 15/12/2018 | 18:00:00 | 62.93 | 0.00 | 0.27 | 0.00 |
| ES | 15/12/2018 | 22:00:00 | 63.43 | -0.02 | 0.28 | 0.02 |
| ES | 16/12/2018 | 02:00:00 | 63.94 | 0.00 | 0.25 | 0.00 |
| ES | 16/12/2018 | 06:00:00 | 64.50 | 0.00 | 0.22 | 0.02 |
| ES | 16/12/2018 | 10:00:00 | 65.40 | -0.04 | 0.20 | 0.01 |
| ES | 16/12/2018 | 14:00:00 | 66.20 | -0.01 | 0.24 | 0.01 |
| ES | 16/12/2018 | 18:00:00 | 66.99 | 0.00 | 0.22 | 0.02 |
| ES | 16/12/2018 | 22:00:00 | 67.92 | -0.05 | 0.27 | 0.01 |
| ES | 17/12/2018 | 02:00:00 | 68.81 | 0.02 | 0.27 | 0.03 |
| LS | 27/2/2019 | 10:00:00 | -70.26 | -2.72 | 0.16 | 0.00 |
| LS | 27/2/2019 | 14:00:00 | -69.99 | -3.82 | 0.18 | 0.01 |
| LS | 27/2/2019 | 18:00:00 | -69.64 | -3.76 | 0.17 | 0.01 |
| LS | 27/2/2019 | 22:00:00 | -68.84 | -3.76 | 0.20 | 0.01 |
| LS | 28/2/2019 | 02:00:00 | -67.99 | -2.96 | 0.30 | 0.01 |
| LS | 28/2/2019 | 06:00:00 | -67.04 | -2.32 | 0.20 | 0.01 |
| LS | 28/2/2019 | 10:00:00 | -65.04 | -1.44 | 0.24 | 0.02 |
| LS | 28/2/2019 | 14:00:00 | -64.94 | -0.66 | 0.16 | 0.02 |
| LS | 28/2/2019 | 18:00:00 | -63.90 | 0.00 | 0.19 | 0.00 |
| LS | 28/2/2019 | 22:00:00 | -62.82 | 0.00 | 0.16 | 0.02 |
| LS | 1/3/2019 | 02:00:00 | -62.18 | 0.00 | 0.18 | 0.00 |
| LS | 1/3/2019 | 06:00:00 | -61.27 | 0.00 | 0.31 | 0.01 |
| LS | 1/3/2019 | 10:00:00 | -60.16 | -0.01 | 0.39 | 0.05 |
| LS | 1/3/2019 | 14:00:00 | -60.01 | -0.86 | 0.28 | 0.02 |
| LS | 1/3/2019 | 18:00:00 | -59.97 | -3.11 | 0.31 | 0.01 |
| LS | 1/3/2019 | 22:00:00 | -59.88 | -5.24 | 0.26 | 0.02 |
| LS | 2/3/2019 | 02:00:00 | -59.86 | -7.35 | 0.36 | 0.02 |
| LS | 2/3/2019 | 06:00:00 | -59.79 | -9.49 | 0.34 | 0.02 |
| LS | 2/3/2019 | 10:00:00 | -59.73 | -11.58 | 0.34 | 0.00 |
| LS | 2/3/2019 | 14:00:00 | -59.74 | -13.69 | 0.35 | 0.02 |
| LS | 2/3/2019 | 18:00:00 | -59.70 | -15.84 | 0.18 | 0.02 |
| LS | 2/3/2019 | 22:00:00 | -59.64 | -17.96 | 0.39 | 0.02 |
| LS | 3/3/2019 | 02:00:00 | -59.62 | -20.08 | 0.25 | 0.01 |
| LS | 3/3/2019 | 06:00:00 | -59.58 | -22.17 | 0.27 | 0.01 |
| LS | 3/3/2019 | 10:00:00 | -59.54 | -24.33 | 0.29 | 0.02 |
| LS | 3/3/2019 | 14:00:00 | -59.48 | -26.34 | 0.30 | 0.01 |
| LS | 3/3/2019 | 22:00:00 | -59.00 | -28.37 | 0.33 | 0.02 |
| LS | 4/3/2019 | 02:00:00 | -58.32 | -29.59 | 0.40 | 0.02 |
| LS | 4/3/2019 | 06:00:00 | -57.62 | -30.83 | 0.30 | 0.03 |
| LS | 4/3/2019 | 10:00:00 | -56.99 | -31.92 | 0.31 | 0.02 |
| LS | 4/3/2019 | 14:00:00 | -56.48 | -32.78 | 0.28 | 0.02 |
| LS | 4/3/2019 | 18:00:00 | -55.96 | -33.65 | 0.31 | 0.01 |
| LS | 4/3/2019 | 22:00:00 | -55.43 | -34.53 | 0.27 | 0.03 |
| LS | 5/3/2019 | 02:00:00 | -54.89 | -35.39 | 0.24 | 0.01 |

| LS | 5/3/2019 | 22:00:00 | -54.18 | -35.46 | 0.34 | 0.02 |
|----|----------|----------|--------|--------|------|------|
| LS | 6/3/2019 | 02:00:00 | -54.17 | -33.59 | 0.31 | 0.02 |
| LS | 6/3/2019 | 06:00:00 | -54.16 | -31.71 | 0.37 | 0.00 |
| LS | 6/3/2019 | 10:00:00 | -54.15 | -29.83 | 0.38 | 0.01 |
| LS | 6/3/2019 | 14:00:00 | -54.11 | -27.89 | 0.24 | 0.01 |
| LS | 6/3/2019 | 18:00:00 | -54.12 | -26.02 | 0.28 | 0.00 |
| LS | 6/3/2019 | 22:00:00 | -54.01 | -24.19 | 0.32 | 0.02 |
| LS | 7/3/2019 | 02:00:00 | -54.12 | -22.83 | 0.32 | 0.04 |
| LS | 7/3/2019 | 06:00:00 | -54.12 | -21.44 | 0.23 | 0.01 |
| LS | 7/3/2019 | 10:00:00 | -54.09 | -19.56 | 0.28 | 0.03 |
| LS | 7/3/2019 | 14:00:00 | -54.09 | -17.59 | 0.29 | 0.01 |
| LS | 7/3/2019 | 18:00:00 | -54.09 | -15.65 | 0.30 | 0.03 |
| LS | 7/3/2019 | 22:00:00 | -54.09 | -13.72 | 0.21 | 0.01 |
| LS | 8/3/2019 | 02:00:00 | -54.06 | -12.25 | 0.21 | 0.04 |
| LS | 8/3/2019 | 06:00:00 | -54.06 | -10.90 | 0.29 | 0.05 |
| LS | 8/3/2019 | 10:00:00 | -54.04 | -8.79 | 0.31 | 0.04 |
| LS | 8/3/2019 | 14:00:00 | -54.04 | -7.01 | 0.49 | 0.02 |
| LS | 8/3/2019 | 18:00:00 | -54.03 | -5.09 | 0.27 | 0.04 |
| LS | 8/3/2019 | 22:00:00 | -54.02 | -3.46 | 0.42 | 0.01 |
| LS | 9/3/2019 | 02:00:00 | -54.01 | -1.82 | 0.32 | 0.01 |
| LS | 9/3/2019 | 06:00:00 | -54.00 | -0.16 | 0.35 | 0.01 |
| LS | 9/3/2019 | 10:00:00 | -53.56 | -0.02 | 0.36 | 0.03 |
| LS | 9/3/2019 | 14:00:00 | -52.78 | 0.00 | 0.26 | 0.01 |
| LS | 9/3/2019 | 18:00:00 | -52.05 | 0.00 | 0.24 | 0.02 |
| LS | 9/3/2019 | 22:00:00 | -51.45 | 0.00 | 0.32 | 0.09 |
| LS | 10/3/2019 | 02:00:00 | -51.01 | -0.22 | 0.27 | 0.03 |
| LS | 10/3/2019 | 06:00:00 | -50.68 | 0.08 | 0.22 | 0.01 |
| LS | 10/3/2019 | 10:00:00 | -50.40 | 1.43 | 0.32 | 0.08 |
| LS | 10/3/2019 | 14:00:00 | -50.15 | 2.85 | 0.25 | 0.02 |
| LS | 10/3/2019 | 18:00:00 | -49.61 | 3.95 | 0.32 | 0.03 |
| LS | 10/3/2019 | 22:00:00 | -48.95 | 4.90 | 0.26 | 0.01 |
| LS | 11/3/2019 | 02:00:00 | -48.23 | 5.82 | 0.36 | 0.09 |
| LS | 11/3/2019 | 06:00:00 | -47.56 | 6.75 | 0.30 | 0.01 |
| LS | 11/3/2019 | 10:00:00 | -46.76 | 7.00 | 0.27 | 0.01 |
| LS | 11/3/2019 | 14:00:00 | -45.87 | 7.06 | 0.34 | 0.04 |
| LS | 11/3/2019 | 18:00:00 | -44.99 | 7.36 | 0.28 | 0.01 |
| LS | 11/3/2019 | 22:00:00 | -44.13 | 7.36 | 0.33 | |
| LS | 12/3/2019 | 00:00:00 | -43.79 | 7.69 | 0.00 | 0.05 |
| LS | 12/3/2019 | 04:00:00 | -43.18 | 7.81 | 0.15 | 0.04 |
| LS | 12/3/2019 | 08:00:00 | -42.89 | 7.01 | 0.22 | 0.00 |
| LS | 12/3/2019 | 12:00:00 | -42.12 | 8.01 | 0.25 | 0.11 |
| LS | 12/3/2019 | 16:00:00 | -41.34 | 9.28 | 0.10 | 0.00 |
| LS | 12/3/2019 | 20:00:00 | -40.54 | 9.94 | 0.02 | 0.01 |
| LS | 13/3/2019 | 00:00:00 | -39.72 | 10.60 | 0.05 | 0.07 |
| LS | 13/3/2019 | 04:00:00 | -38.89 | 11.25 | 0.01 | 0.01 |
| LS | 13/3/2019 | 08:00:00 | -38.06 | 11.92 | 0.01 | 0.01 |

| | | | | | | |
|---|---|---|---|---|---|---|
| LS | 13/3/2019 | 12:00:00 | -37.24 | 12.56 | 0.01 | 0.01 |
| LS | 13/3/2019 | 16:00:00 | -36.42 | 13.19 | 0.02 | 0.01 |
| LS | 13/3/2019 | 20:00:00 | -35.58 | 13.83 | 0.01 | 0.01 |
| LS | 14/3/2019 | 00:00:00 | -34.92 | 14.32 | 0.00 | 0.00 |
| LS | 14/3/2019 | 04:00:00 | -34.50 | 14.87 | 0.01 | 0.01 |

Lines 219 and 222 – The authors should state the calculated p-values, and not just state that the values are or aren't significant. Moreover, are p-values really appropriate for this data set? I encourage the authors to think about this article: https://link.springer.com/article/10.1007/s10654-016-0149-3. Perhaps a simple comparison of the data would be more insightful than a "significant/not significant" binary outlook.

*As per the reviewer's suggestion, we will include the calculated p-values in the text as opposed to simply stating whether values are or aren't significant. The conclusions drawn about the difference in the $\delta^{18}O$ data in the Weddell Sea vs. the latitudinal transect are primarily based on the spatial comparison (i.e., Figs 1B and 3). The statistical significance was added as a supporting note. We agree with the reviewer that relying on the significance of p-values to drive comparisons is not always valid, but still feel that it is relevant to keep the information in the manuscript with a more precise p-value provided.*

Figure 1 – the points in this figure are rather coarse, and it might be easier to see the points and error bars with thinner lines for both.

*The linewidth of the symbols and error bars will be reduced by half to make the points clearer.*

Figure 2 - It might make it somewhat more complicated, but it would be helpful to, rather than colour all the back trajectory lines the same tone of grey, to have them coloured by the date of the sample, even if it were done using an ombre (monotone) effect. In particular, this would be helpful for the Weddell Sea legs, because there is a significant amount of overlap in the back trajectories. Alternatively, individual or grouped back trajectories could be shown in a Supplemental Figure, to complement the complete regional back-trajectory version that is in this figure. Also, "AMBTs" doesn't need an apostrophe in the figure caption (and in the caption for Figure 3.)

*The apostrophe in AMBTs will be removed from the figure captions of Figures 2 and 3.*
*We agree with the reviewer that there is significant overlap in AMBTs from the Weddell Sea, which makes it difficult to see exactly where the air from each separate filter deployment originated from. This is much clearer in the latitudinal early and later summer cruise transects as the filter deployments are spaced out. For interested readers we will include a supplementary figure that shows the AMBTs for every filter deployment from the Weddell Sea. This additional figure is shown below (Figure SX).*

[Figure]

[Figure]

[Figure]

*Figure SX. Maps depicting 72-hour AMBTs computed each hour for every filter deployment (dark grey dots) during the Weddell Sea leg of the research voyage. Each subplot represents one filter deployment, and the cruise leg ID is shown (Tables S1 and S3) in the legend on the upper right-hand side of each subplot. The red dots highlight the ships path.*

Figure 4 - the back trajectory lines are again quite thick – if possible, it might be better to show the back trajectory lines with a slightly finer point size. Also, in the caption, lines 304-305 – "AMBTs" x2. Line 306 – "sea ice" (ice is a noun), and in line 307 "sea-ice concentration" (sea-ice is an adjective.) As well, "AMSR2 ASI programme" needs to be defined, either here or in the main text. Finally, "the white region represents the location…" of the sea ice identified by the AMSR2 ASI programme in the Antarctic study region, not "at the southernmost extent of each transect".

*The apostrophe will be removed from both AMBTs instances.*
*"Sea ice" will be used everywhere in the caption when it is a noun, and "sea-ice" will be used when it is an adjective. The caption will be edited to reflect that the white shows the location of sea ice determined from satellite derived sea-ice concentration data obtained from the passive microwave sensor, AMSR2. AMSR2 will be described in the figure caption as "Advanced Microwave Scanning Radiometer 2, (Spreen et al., 2008)". The trajectory lines will also be plotted with a finer point to make them more visible.*

Line 310 – many of the transects have two shades of orange, indicating that they originated in one region, and then continued through a different region before being sampled. Were these samples treated differently in your statistics than the ones that were modeled to be entirely in one region throughout the entire 72 hours? This needs to be clarified.

*The AMBTs of all the samples originated from at least 2 zones, therefore their $\delta^{15}N$-$NO_3^-$ signature is derived from at least 2 different $NO_x$ sources in all cases. There were no samples that were assumed to be entirely in one region throughout the entire 72 hours.*
*For each filter deployment (i.e., each sample), a 72-hour AMBT was run every hour. Each 72-hour AMBT for the whole filter deployment was taken into consideration when determining how much time was spent in each zone. This is clarified in lines 352-356 where the AMBTs are used along with the source $NO_x$ $\delta^{15}N$ values to derive the estimated alkyl nitrate influenced signal. The lack of clarity is perhaps due to the citation of figure 4 in line 310. We have amended the text so figure 2 is cited in line 310, which is more useful for evaluating the sentence "At the northern extent of our transects, the low-latitude aerosol samples, defined as those originating from anywhere north of 43°S in early summer and 41°S in late summer (Figure 2), had the highest average $\delta^{15}N$-$NO_3^-$ signature (-4.9 ±*

*1.5‰; n = 5).” As a result, the current Figure 4 (which will now be figure 5) will not be introduced until line 349 when the discussion of the use of AMBT to assist with determining the regions that contributed to the signal for each sample. Here we will also amend the text to clarify what the colours of the AMBTs in current Figure 4 represent. The text to be included is as follows: “We split the latitudinal transect into three regions, each characterised by the dominance of a different natural source of $NO_3^-$ i.e., lightning $NO_x$ at the low-latitudes (Figure 5 light orange), oceanic $RONO_2$ emissions at the mid-latitudes (Figure 5 dark orange) and snowpack emissions at the high-latitudes (Figure 5 red).”*

*In addition we have edited figures 2 and 3 to include the location of the sea ice.*

Lines 318-319 – “end of the low-latitude zone” – this is still somewhat ambiguous.

*The start of the mid-latitude zone which is also the end of the low latitudes zone was identified as the latitude at which non-zero sea surface nitrite concentrations begin to become prominent. We will correct this by explicitly stating these latitudes as 43°S and 41°S in early and late summer, respectively to make it clearer. In addition, we will delete the clause “i.e., end of the low latitude zone”. The sentence will now read:” The beginning of the mid-latitude zone (i.e., 43°S and 41°S in early and late summer, respectively) was defined by the presence of non-zero sea surface nitrite concentrations in early and late summer (Figure 4).”*

Lines 363-370 – While I certainly appreciate the methodology being used here, there should be a discussion about the uncertainties inherent from putting so much of the analysis on the accuracy of the back trajectories. There needs to be a discussion about the reliability and uncertainties in the HYSPLIT AMBTs, and the resultant uncertainties in the calculated isotopic impacts on δ15N of aerosol nitrate.

*We agree with the reviewer that a discussion of the uncertainty/reliability of HYSPLIT in producing AMBTs is useful considering the extent to which we rely on AMBT information to estimate the isotopic signature of aerosol nitrate derived from oceanic alkyl nitrates.*

*We are aware that HYSPLIT has limitations owing to multiple factors including the spatial and temporal resolution of the meteorological data used to force the model. The longer the back trajectory is, the less spatial coherency is observed, due to the propagation of error along back trajectories (Sinclair, et al., 2013). This can impact the accuracy of the location of each AMBT computed. According to literature the spatial uncertainty is estimated to be 15% to 30% of the travel distance because of errors in wind fields (Sarchilli et al., 2011).*

*Nevertheless, HYSPLIT is a frequently used tool for assessing air mass origin in the Southern Hemisphere and over Antarctica and is used in all three of the most cited papers for comparison purposes within this manuscript (Morin et al., 2008; Walters et al., 2019; Shi et al., 2021).*

*Perhaps most importantly with respect to this study, knowing the location of each AMBT with exact precision is not necessary given that we are operating over such large spatial scales. For example, if air masses originate from anywhere north of 43° S they are considered to be influenced by predominantly lightning $NO_x$. Variation in the path of an AMBT within a zone will not change the percent contribution of the dominant $NO_x$ source in that zone to $NO_3^-$, or by extension the $\delta^{15}N$ value that we estimate for $NO_3^-$ derived from $RONO_2$ emissions. The accuracy of AMBTs is perhaps more influential at the high latitudes where we assess contact with sea/continental ice. Even then, if we*

*simply assumed a threshold latitude and suggested that every AMBT located south of 70º is predominantly influenced by snow $NO_x$ emissions, this would not change our results significantly (the $\delta^{15}N\text{-}NO_3^-$ originating from $RONO_2$ emissions in this case would be -23.5‰ as opposed to -22‰). However, the northern edge of the sea ice is not uniform with longitude, thus we used the AMBT locations that corresponded with locations of > 50% sea ice coverage in an attempt to be more accurate.*

*Furthermore, our AMBTs are short (72 hours), which alleviates some of the uncertainty caused by the propagation of error along back trajectories, as compared to studies with AMBTs of >5 days. We will add the following to the text to clarify the potential uncertainty associated with the AMBTs, and that this uncertainty is unlikely to influence the estimated $\delta^{15}N$ end member associated with alkyl nitrate emissions.*

*Text to be included at line 367: "Using this approach to estimate the $\delta^{15}N\text{-}NO_3^-$ from oceanic $RONO_2$ emissions relies heavily on AMBTs generated using HYSPLIT. While HYSPLIT is a frequently used tool for assessing air mass origin in the Southern Hemisphere and over Antarctica (Morin et al., 2008; Walters et al., 2019; Shi et al., 2021), it is important to note that a spatial uncertainty of 15% to 30% of the trajectory path distance can be expected (Sarchilli et al., 2011). AMBTs also become increasingly uncertain the further back in time they are used (Sinclair, et al., 2013). Some of this uncertainty is alleviated by the fact that the AMBTs generated here are relatively short. Additionally, the spatial scale of the low-, mid- and high-latitude zones is large, such that some variation in sample AMBTs will not significantly alter the samples dominant $NO_3^-$ source."*

Technical comments

Title – The period at the end of the title is unnecessary

*The period at the end of the title will be removed.*

Line 31 – here and throughout the paper, per the EGU style guide, use "and" instead of &, both in in-text citations and in the reference list. Similarly, per the style guide, for Figure panel labels, use lower case letters, i.e., (a), (b), etc. Also, "Coordinates need a degree sign and a space when naming the direction (e.g. 30° N, 25° E)", and "Common abbreviations to be applied: hour as h (not hr), kilometre as km, metre as m". Also, Figure captions should be numbered "Figure 1…", not "Fig. 1…", and Figures, Equations, and Sections should be referred to as "Fig. #, "Eq. (#)", and "Sect. #" when not at the beginning of a sentence. Likewise, reactions should be referenced in the text in parentheses: e.g., (R10).

*We thank the reviewer for this technical guidance and will be sure to follow the EGU style guide by converting all "&" to "and" in both in-text citations and the reference list. We will also change figure panel labels to be in lower case letters. We will edit the co-ordinates to include a degree sign and a space. Lastly we will edit all figure captions to be numbered using "Figure X" as opposed to "Fig. X" and all reactions in the text will be correctly referenced using parentheses.*

Line 31 – "Earth's"

*This will be edited.*

Line 38 – probably out to put "Southern Ocean (SO)" here, so later references to SO are defined.

*"Southern Ocean (SO)" will be included.*

Line 39 and throughout – references with "et al., YYYY" should not have a comma following the first author's last name.

*Any case of a comma following the first authors last name will be removed in references with "et al., YYYY"*

Line 53 – "(Jones et al., 2000, 2001)."

*This will be edited as per the reviewer's suggestion.*

Line 80 – remove the word "both" (three things are listed, so "both" doesn't make sense".)

*The word "both" will be removed.*

Line 86 – "R" should be italicized.

*R will be italicized.*

Line 94 – "(Berhanu et al., 2014, 2015)".

*This will be edited as per the reviewer's suggestion.*

Line 179 – Probably ideal to use the same notation for pooled standard deviation here and in Table S2, either sp (with subscripted p), or SDp (with a subscripted p.)

*We will correct this by using same notation for pooled standard deviation here and in Table S2.*

Line 190 – "BÓ§hlke et al."

*This will be edited as per the reviewer's suggestion.*

Line 198 – "ship's"

*This will be edited as per the reviewer's suggestion.*

Line 201 – This should probably be section 2.3, not 2.6. Also, the numbering notation of the section notation should be consistent throughout the manuscript for each type of heading: 1), 2), etc., or 1.1 Secondary Heading, 2.2 Another Secondary Heading, etc.

*We will change this to refer to section 2.3 as opposed to 2.6 and ensure that our numbering notation for sections is consistent.*

Line 207 (and Line 699) – I believe it is https://

*This will be edited*

Line 214 – "high latitudes"

*This will be edited*

Line 228 (Table 1) – $N_2$ should have a subscripted "2".
*The subscript "2" will be included.*

Lines 228 (Table 1) and 230 (Figure 1 caption) – The convention for the standard notation, as you have on Line 87 and Table S2, is "VSMOW", not "V-SMOW".

*The convention "VSMOW" will be used.*

Line 241 – "Our observations reveal…" would be sufficient.

*This will be edited.*

Line 247 – "AMBTs"

*This will be edited.*

Lines 265-266 – pptv and ppbv should be defined.

*We now only refer to ppbv in the text, and will define this as parts per billion by volume.*

Line 287 – "high latitudes"
*This will be edited.*

Lines 323, 324, 360 – "AMBTs"
*This will be edited.*

Line 329 – "Dahl and Saltzman, 2008;".
*This will be edited.*

Line 342 – "NOx" should have a subscripted x.

*This subscript x will be included.*

Line 351 – There should be a comma before "i.e.,"
*A comma will be added before "i.e.,"*

Lines 379-380 – recommend italicizing "*f*", here and later.

*We will italicize "f" here and elsewhere*.

Line 383 and 384 – add a comma prior to i.e.
*A comma will be added before "i.e.,"*

Line 404 – "hypothesize" (or hypothesise for regional spelling consistency)
*This will be edited.*

Line 418 – HCl should have a lower case L.

*A lower case "l" will be included here.*
Line 449 – "AMBTs", and the light blue lines aren't dashed.
*This will be edited.*
Line 454 – " for 15 January 2019".
*This will be edited.*
Line 478 – "At the mid-latitudes, peroxy…" and "while in the"
Line 507 – "J.G." (for consistency with other referenced (co-)authors.)
*This will be edited.*
Line 514 – use https://doi.org... formatting throughout the references. Also, here and throughout the reference list, per the EGU ACP guide, Journal Abbreviations should be used.
*https://doi.org formatting will be used throughout the reference list and journal abbreviations will be used.*

Line 527 – "Journal of Geophysical Research Letters" should be "J. Geophys. Res. - Atmos.", and "Ocean" should be capitalized.

*This will be edited.*

Line 545 – C1 and C4 should have 1 and 4 subscripted. Also, "Letters" on the next line should also be "J. Geophys. Res. - Atmos."
*This will be edited.*

Line 550 – also JGR-A, not GRL. And the DOI link is https://doi.org/10.1029/1999JD900238.
*This will be edited.*

Line 573 – "Saltzman, S. E."
*This will be edited.*

Line 575 – "Craig", not "Graig"; this article is the ACP version, so remove "Discussions" from line 577.
*This will be edited and "Discussions" will be removed.*

Line 579 – DOI citation: https://doi.org/10.1023/A:1009738715891
*This will be included.*
Line 580 – This should be "Elliott, E. M." Also, in "United States," (there is a space and a period where there should be a comma… likewise, in several other references, there is a period instead of a comma following the article title.)
*This will be edited.*

Line 587 – "nitrogen"
*This will be edited.*

Line 599 – there is a rogue comma detached from "Meteorology".
*This will be edited.*

Line 601 – "… Research: Atmospheres", DOI: https://doi.org/10.1029/93JD00874
*This will be included.*
Line 606 – "aerosol"
*This will be edited.*

Line 612 – "Atmospheric" is spelled incorrectly (although it should be "Atmos. …" and this is not a discussion paper, so remove "Discussions" from line 613.
*"Discussions" will be removed and Atmospheric abbreviated.*
Line 614 – if this is a book, it should have more details.
*The full details will be included.*

Line 619 – there is a space missing in "… vapor at…", Maido should be "Maïdo", and the DOI citation is: https://doi.org/10.1002/2017JD026791
*This reference will be corrected and DOI included.*
Line 628 - DOI citation: https://doi.org/10.1029/1999RG000078
*This DOI will be included.*

Line 630 – DOI citation: https://doi.org/10.1016/0016-7037(57)90021-2
*This DOI will be included.*

Line 643 – "Peroxyacetyl"
*This will be edited.*

Line 649 – The date for this citation (https://doi.org/10.1029/97JD02075) is in the wrong location.
*The location of the date will be edited.*
Line 655 – in the reference, it is spelled "Oxidized". And "Atmos." is missing from the Journal title.
*This will be edited.*

Line 660 – "formation"
*This will be edited.*

Line 674 – "Lee, H.-M., …" – the H is missing a period.
*A period will be included.*

Line 684 – "Atmos." is missing.
*"Atmos." Will be included.*

Line 687 – "Müller"
*This will be edited.*

Line 691 – "NOx" should have a subscripted x.
*The x will be subscripted.*

Line 705 – remove "Discussions."
*Discussions will be removed.*

Line 715 – "Galanter, M. and…"
*This will be edited.*

Line 727 – "Atmos." is missing.
*"Atmos." Will be included.*

Line 747 – "Atmos." is missing.
*"Atmos." Will be included.*

Line 754 – "Comparisons" is spelled incorrectly, and it should be "Atmospheric Environment" (no 's'), but of course, "Atmos. Environ."

*Spelling will be corrected, and the abbreviated journal name will be used.*

Line 754 – "NOx" should have a subscripted x.
*The x will be subscripted.*

---

## Author Comment (AC2)

General Comments

In this paper the authors present a very interesting and unique set of data on the isotopic composition of coarse mode atmospheric nitrate in the Southern Ocean marine boundary layer. The data were derived from latitudinal transects as well as sampling near the Weddell Sea and Ronne-Filchner Ice Shelf. The authors combined their isotopic measurements of aerosol nitrate with air-mass back trajectory (AMBT) analysis to determine the major sources of NOx contributing to nitrate aerosols in the Southern Ocean. The paper is well written, the data are clearly presented, and the conclusions derived from the study are well supported, with the possible exception of the estimated δ15N signature of oceanic nitrate emissions (see below). I discuss a couple of potential technical issues with the isotope measurements below. I also think that the authors can do more with their dataset – it seems a little over aggregated. For example, I am curious to know how the relationship between δ18O and δ15N varies in the study and if the different size fractions of aerosol had different N masses and isotopic values.

*We thank the reviewer for their constructive comments and support for the manuscript. Below, we address each of the reviewer's specific and technical line-by-line comments. The reviewer comments are in black text while the responses are in blue italics.*

Specific Comments:
Lines 88-131. This is a very nice introduction to the main reactions affecting the isotopic composition of oxidized N in the atmosphere.

*Thank you, we appreciate the reviewers' careful read of the manuscript.*

Line 150-151. State the specific Tisch Environmental filter used along with its surface area.
*As per the reviewer's suggestion, the specific filters used (TE-230-GF; Tisch Environmental) will be stated along with their surface area (119 cm$^2$).*

Line 171. Ultra-clean deionized water is vague. Please state the resistance of the DI water in megaohms.

*The resistance of the DI water (18.2 MΩ) will now be stated in the manuscript.*

Lines 178-179. I am confused by the presentation of the filter blanks. The text states that the average amount of nitrate on a field-blank filter was 484.7 nmoles or 0.4847 μmoles (also, this value should have a standard deviation accompany it). Since the filters were extracted in 25 ml of DI water, 0.4847 μmoles of nitrate would result in a nitrate concentration of 19.4 μmoles/liter which seems very high for a blank. I suspect that only a fraction of the blank filter was extracted in DI (hence the approximately 30 cm2 reference in line 172) and if this is the case it should be included in the methods section to avoid confusion.

*We agree with the reviewer that the presentation of the filter blanks is confusing in its current form. To clarify, the reviewer is correct in assuming that a fraction of each filter blank (30 cm$^2$) was extracted in 30 mL of DI. For a clearer presentation of the filter blanks, we will instead refer to the blank as a percentage of the total concentration as follows: "Final aerosol [NO$_3^-$] were corrected by subtracting the field blanks, which typically represented 35% of total [NO$_3^-$] on average."*

Later the authors state that "The pooled standard deviation (Sp) of four repeated sample measurements for [NO3-] was 0.3 μmol L-1". Does this sentence refer to an actual aerosol sample or are the authors referring to the blank filters? If the sentence refers to blank filters, then it seems that only four blanks were run over several weeks of aerosol sampling. For context, the authors should state the range of nitrate values measured in the actual aerosol sample extracts.

*The pooled standard deviation (Sp) of repeated sample measurements for nitrate refers to 4 actual aerosol samples that were measured in duplicate during the IC run. This was included to give an example of instrument precision for the IC system. As such, we will refer to this value (0.3 µmol L$^{-1}$) as the precision of the instrument as opposed to the pooled standard deviation of repeated sample measurements to avoid confusion.*

*We will also include the range of nitrate values measured in the actual aerosol sample extracts (1.3 to 27.7 µmol/L) for context, as per the reviewer's suggestion.*

Lines 188-190. I believe it is more common for investigators to use the USGS 32 isotope standard (+180 per mil) and USGS 34 (-1.8 per mil) for calibration of δ15N values measured with the microbial denitrifier method. In the present study IAEA N3 (+4.7) was substituted for USGS 32 resulting in a very truncated isotopic baseline for calibration. I wonder if the authors could explain why they chose N3 over USGS 32 and if they think their δ15N values are comparable to those in other papers.

*While it is true that some studies use USGS32 in addition to USGS34 or IAEAN3, the literature shows a range of isotope standards used for sample correction. Ideally, the standards used (whether it is 2 or 3) have isotopic values that are similar to the upper and lower bounds of the samples analyzed. The dynamic range we observe in δ$^{15}$N-NO$_3^-$ is -43.1 to 2.7‰. Unfortunately, there is no isotopic standard with a low δ$^{15}$N comparable to the negative values observed in this dataset. Given the values observed here, the use of a very high δ$^{15}$N standard such as USGS32 would possibly result in a calibration line whose slope was dictated primarily by an isotopic value that is ~ 220‰ away from the lowest observed value. The slope of the correction line would be heavily influenced by the accuracy of the USGS32 analysis, which would introduce error unnecessarily. In the end, the reference materials used here still make our data comparable with other studies, particularly atmospheric studies.*

Lines 195-197. For each filter deployment, the authors weighted the isotopic values of the size fractions based on the mass of nitrate in the fractions. This seems very reasonable and it simplifies the data presentation. However, I am curious if there were any consistent mass or isotopic patterns in the size fractions. I believe a small table and a few lines of text would be quite valuable to readers.

*We did not include figures showing the nitrate concentration and isotopic composition of nitrate in each size fraction because there were not clear trends with size. We will however include a table of NO$_3^-$ concentration and δ$^{15}$N-NO$_3^-$ per size fraction in the supplemental material for comparison with other size-segregated aerosol studies. The table that we will include is shown below (Table SX).*

*Table SX: The average (Avg) and standard deviation (SD) for atmospheric nitrate concentration ([NO$_3^-$] (ng m$^{-3}$)) and nitrogen isotopic composition (δ$^{15}$N-NO$_3^-$) in each aerosol size range: >7 µm, 3 to 7 µm, 1.5 to 3 µm and 1 to 1.5 µm, separated by cruise leg: early summer (ES), Weddell Sea (WS) and late summer (LS).*

| | [NO$_3^-$] (ng m$^{-3}$) | | | | | | δ$^{15}$N-NO$_3^-$ (‰ vs. N$_2$) | | | | | |
| | ES Avg (SD) | n | WS Avg (SD) | n | LS Avg (SD) | n | ES Avg (SD) | | WS Avg (SD) | n | LS Avg (SD) | n |
|---|---|---|---|---|---|---|---|---|---|---|---|---|
| **>7 µm** | 10.7 (17.4) | 7 | 4.1 (4.3) | 20 | 6.9 (6.0) | 7 | -12.2 (11.2) | 6 | -14.6 (5.1) | 20 | -10.4 (2.7) | 7 |
| **3 to 7 µm** | 29.9 (28.9) | 7 | 7.7 (4.9) | 20 | 23.3 (25.7) | 7 | -18.8 (14.9) | 7 | -25.8 (9.0) | 20 | -13.7 (6.0) | 7 |
| **1.5 to 3 µm** | 27.1 (19.3) | 7 | 9.4 (3.6) | 20 | 17.9 (17.7) | 7 | -20.1 (16.5) | 7 | -24.5 (6.6) | 20 | -15.9 (8.3) | 7 |
| **1 to 1.5 µm** | 20.5 (6.4) | 7 | 8.3 (3.4) | 20 | 11.6 (6.0) | 7 | -19.7 (15.5) | 7 | -23.0 (7.4) | 20 | -16.7 (9.5) | 7 |

Lines 198-200. Why were only nitrite concentrations measured in seawater (what about nitrate)? Also, it seems like isotopic measurements of ocean water NO2 and or NO3 would be useful for the later discussion surrounding RONO2 as a source of atmospheric N.

*Both seawater nitrate and nitrite concentration measurements were conducted during the research voyage. We only report on surface ocean nitrite concentrations in this manuscript as that is the precursor for alkyl nitrate formation. Modelling studies suggest that oceanic alkyl nitrates can only occur in regions of the ocean that possess non-zero sea surface nitrite concentrations (Fisher et al., 2018). Theory surrounding the aqueous phase chemical formation of alkyl nitrates in the surface ocean suggests no role for sea surface nitrate, therefore it's concentration and $\delta^{15}N$ are not relevant here.*

*The N isotopic composition of alkyl nitrates in the surface ocean has never been measured, however the $\delta^{15}N$ of oceanic nitrite can be very low in the Atlantic sector of the Southern Ocean, ranging from -20 to -60‰ (Fripiat et al.,2019), as seen in the figure below taken from Fripiat et al., 2019.*

[Figure]

*One might assume that alkyl nitrates will have a very low $\delta^{15}N$ signature as a result. However, to our knowledge there are no studies that investigate the nitrogen isotope fractionation during 1) alkyl nitrate formation in the surface ocean, 2) the flux of alkyl nitrates out of the ocean, and/or 3) the formation of aerosol nitrate from atmospheric alkyl nitrates. As a result, we cannot comment on the extent to which the isotopic composition of alkyl nitrates will reflect the isotopic composition of sea surface nitrite, and this is why we did not include mention of the Fripiat study in our paper.*

*It should also be noted that the estimated $\delta^{15}N$ signature that we report for atmospheric nitrate, originating from surface ocean alkyl nitrates, is not necessarily an estimate of the $\delta^{15}N$ signature of the alkyl nitrate source itself. We address this issue further in the reviewers last point below and detail modifications made to the text to clarify these points.*

Lines 217-219. State the latitudinal range of values used in the comparison of Weddell Sea and transect samples.

*The latitudinal range of values used in comparison of Weddell Sea and transect samples will be included. The latitudinal range was 56.0 ºS to 70.2 ºS.*

Line 229/Figure 1. Figure 1 is a good overview of the data, but I was curious about how these relationships varied when the data were disaggregated. Fortunately the authors provided a very nice table of data in the Supplemental Materials. Here are some additional graphs that I made with the data that bolster several of the authors hypotheses in the Discussion.

*We thank the reviewer for their positive comments on Figure 1 and the supplementary material, as well as their use of figures in support of our hypotheses. We have decided to include the cross plot of atmospheric [NO₃⁻] and $\delta^{15}N$-NO₃⁻ as a supplementary figure introduced at line 312, to support our hypothesis that the latitudinal gradient in lightning generated $NO_x$ leads to higher [NO₃⁻] and higher $\delta^{15}N$-NO₃⁻ at the low latitudes. The amended text will read: "Lightning activity at the low latitudes is*

*also consistent with the higher atmospheric [NO₃⁻] observed (Fig. 1A) and is further supported by co-occurring high [NO₃⁻] and relatively high δ¹⁵N-NO₃⁻ values (Fig. SX)."*

[Figure]

*Figure SX. The average (± 1 SD) coarse mode (> 1 μm) nitrate concentration [NO₃⁻] (ng m⁻³), plotted as a function of the weighted average (± 1 SD) δ¹⁵N-NO₃⁻ (‰ vs. N₂). Early and late summer latitudinal transects are denoted by the red triangles and green squares, respectively. Weddell Sea samples are denoted by blue circles. Where error bars (± 1 SD) are not visible, the standard deviation is smaller than the size of the marker.*

Lines 243, 285 and elsewhere. Try to avoid the use of "very". When stating something is high or low it is probably better to say "relatively high" or "relatively low" based on your ranges of data values.

*As per the reviewer's suggestion, we will avoid using the word "very" when trying to state that something is high/low.*

Line 250. It might be useful for readers if you gave a range of NO3 concentrations in urban airsheds for purpose of comparison.

*As per the reviewer's suggestion, we will include a range of NO₃⁻ concentrations typical of urban airsheds for comparison purposes. The first paragraph of section 4.1 will be edited as follows:*

*Aerosol NO₃⁻ concentrations were low (< 100 ng m-3; Fig. 1A) for most air masses sampled along the latitudinal transect and in the Weddell Sea, consistent with the expectation of minimal influence from anthropogenic NOₓ sources. For comparison, NO₃⁻ concentrations in a polluted urban airshed over South Africa can be > 500 ng m⁻³ (Collett et al., 2010).*

Line 255. How is lifetime defined?

*Lifetime is defined as the amount of time the species (in this case NO₃⁻) remains in the atmosphere before being removed.*

Lines 265-266. Keeping the units the same might make this comparison clearer.

*The units will be kept the same to make the comparison clearer. The sentence will read: "Typical O₃ concentrations observed at coastal sites in Antarctica are on the order of 20 ppbv (Nadzir, et al., 2018), whereas the sum of NO and NO₂ rarely exceeds 0.04 ppbv (Jones, et al., 2000; Weller, et al., 2002; Bauguitte, et al., 2012).*

Lines 260-274. The arguments against equilibrium fractionation having a role in the isotopic dynamics of atmospheric NOx are logical and convincing.

*We thank the reviewer for their agreement with our argument in this case.*

Line 281. Delete word "therefore".

*The word "therefore" will be deleted.*

Line 286. I suggest changing "lightning" to "thunderstorms" and "this mechanism" to "lightning".

*Here we incorrectly reference (Savarino et al., 2007), which makes the sentence unclear. A more appropriate reference (Nesbitt et al., 2016) will be included in the manuscript and line 286 will read: "The latitudinal gradient in lightning NOₓ production suggests that lightning NOₓ is greatly reduced at high-latitudes (Nesbitt et al., 2016)."*

*Nesbitt et al., (2016) report global and seasonal distributions of NOx production by lightning using data from the Optical Transient Detector (OTD), a space-borne lightning sensor that detects lightning flashes. We prefer not to use the word thunderstorms in place of lightning as to avoid confusion, given that some studies assess lightning produced NOₓ using the observed distribution of electrical storms and cloud characteristics. In this case we believe using the word "lightning" is more accurate.*

Line 299. Remove word "previously".

*The word previously will be removed as per the reviewer's request.*

Line 317. Suggest changing "unlikely to be influenced" to "likely to be less influenced".

*We thank the reviewer for the suggestion but disagree with respect to this change. It will alter the meaning of the sentence and will suggest that snow NOx is potentially important, which we are explicitly excluding with this sentence. Please also see below.*

Section 4.2.2. Based solely on the stable isotope data presented, there does not need to be more than two endmembers to explain the range of δ15N values observed in the study: i) near zero per mil lightning source and ii) relatively low, -48 to -56 per mil NO3 produced from photolysis from snow/ice. However, the authors have combined the isotope data with AMBT to gain better understanding of the atmospheric dynamics during their cruise. This analysis suggests that ocean-derived NOx, with intermediate δ15N (~-22 per mil), could be an important atmospheric source in the mid-latitude samples. They then hypothesize that RONO2 emissions are a major source of NOx in the Southern Ocean MBL.

I think this hypothesis needs some beefing up. First, the authors should discuss the accuracy of the AMBTs – are the models reliable in the Southern Ocean? Second, I think the authors should describe the biotic and abiotic processes that generate oceanic RONO2 and the substrates on which these reactions rely. Third, it would be very useful to discuss both the range of δ15N of the substrates (NO2

and NO3?) as well as the direction and magnitude of the N fractionations resulting from the production of oceanic RONO2. While the authors did not measure δ15N of NO2 and NO3 in their study, there are likely other data available from the Southern Ocean to include. Basically, what I'd like to see is if it is reasonable for RONO2 to have a δ15N of -22 +/- 7.5 per mil based on what we know about the substrates and reactions that produce it.

*We thank the reviewer for their suggestions to strengthen our hypothesis regarding the oceanic RONO$_2$ source. To address the reviewers comments we will first discuss the accuracy of HYSPLIT modelled AMBTs in the Southern Ocean.*

*We agree with the reviewer that a discussion of the uncertainty/reliability of HYSPLIT in producing AMBTs is useful considering the extent to which we rely on this AMBTs information to estimate the isotopic signature of aerosol nitrate derived from oceanic alkyl nitrates.*

*We are aware that HYSPLIT has limitations owing to multiple factors including the spatial and temporal resolutions of the meteorological data used to force the model. The longer the back trajectory is, the less spatial coherency is observed, due to the propagation of error along back trajectories (Sinclair, et al., 2013). This can impact the accuracy of the location of each AMBT computed. According to the literature the spatial uncertainty is estimated to be 15% to 30% of the travel distance because of errors in wind fields (Sarchilli et al., 2011).*

*Nevertheless, HYSPLIT is a frequently used tool for assessing air mass origin in the Southern Hemisphere and over Antarctica and is used in all three of the most cited papers for comparison purposes within this manuscript (Morin et al., 2008; Walters et al., 2019; Shi et al., 2021).*

*Perhaps most importantly with respect to this study, knowing the location of each AMBT with exact precision is not necessary given that we are operating over such large spatial scales. For example, if air masses originate from anywhere north of 43º S they are considered to be influenced by predominantly lightning NO$_x$. Variation in the path of an AMBT within a zone will not change the percent contribution of the dominant NO$_x$ source in that zone to NO$_3^-$, or by extension the δ$^{15}$N value that we estimate for NO$_3^-$ derived from RONO$_2$ emissions. The accuracy of AMBTs is perhaps more influential at the high latitudes where we assess contact with sea/continental ice. Even then, if we simply assumed a threshold latitude and suggested that every AMBT located south of 70º is predominantly influenced by snow NO$_x$ emissions, this would not change our results significantly (the δ$^{15}$N-NO$_3^-$ originating from RONO$_2$ emissions in this case would be -23.5‰ as opposed to -22‰). However, the northern edge of the sea ice is not uniform with longitude, thus we used the AMBT locations that corresponded with locations of >50% sea ice coverage in an attempt to be more accurate.*

*Furthermore, our AMBTs are short (72 hours), which alleviates some of the uncertainty caused by the propagation of error along back trajectories, as compared to studies with AMBTs of >5 days. We will add the following to the text to clarify the potential uncertainty associated with the AMBTs, and that this uncertainty is unlikely to influence the estimated δ$^{15}$N end member associated with alkyl nitrate emissions: "Using this approach to estimate the δ$^{15}$N-NO$_3^-$ from oceanic RONO$_2$ emissions relies heavily on AMBTs generated using HYSPLIT. While HYSPLIT is a frequently used tool for assessing air mass origin in the Southern Hemisphere and over Antarctica (Morin et al., 2008; Walters et al., 2019; Shi et al., 2021), it is important to note that a spatial uncertainty of 15% to 30% of the trajectory path distance can be expected (Sarchilli et al., 2011). AMBTs also become increasingly uncertain the further back in time they are used (Sinclair, et al., 2013). Some of this uncertainty is alleviated by the fact that the AMBTs generated here are relatively short. Additionally, the spatial scale of the low-, mid- and high-latitude zones is large, such that some variation in sample AMBTs will not significantly alter the expected dominant NO$_3^-$ source."*

*Second, a description of reactions leading to oceanic RONO$_2$ including the substrates of which these rely, are outlined below. In the introduction we outline that although the exact mechanism remains*

*unclear, experimental evidence suggests that oceanic RONO$_2$ production occurs via photochemical processes involving the aqueous phase reaction of RO$_2$, derived from the photolysis of oceanic dissolved organic matter and NO, derived from seawater nitrite photolysis (Dahl, et al., 2003; Dahl & Saltzman, 2008).*

$NO_2^- \xrightarrow{hv, H_2O} NO + OH + OH^-$ *(R1)*

$CDOM \xrightarrow{hv, O_2} ROO$ *(R2)*

$ROO + NO \rightarrow RONO_2$ *(R3)*

*The alkyl nitrate then fluxes out of the ocean into the atmosphere, and either undergoes hydrolysis to form aerosol nitrate or is photolyzed to NO$_x$, which subsequently forms aerosol nitrate. The $\delta^{15}N$ signature of alkyl nitrate is unknown, but the $\delta^{15}N$-NO$_2^-$ measured in the Southern Ocean is very low (Fripiat et al. 2019). However, as mentioned in response to the reviewer above, we do not know the extent to which the $\delta^{15}N$ signature of nitrite is conserved during the various processes between its formation in the surface ocean and removal as aerosol nitrate. Therefore, based on our data we cannot comment on the $\delta^{15}N$ of alkyl nitrate itself, or if the value we estimate is consistent with expectations based on surface ocean nitrite $\delta^{15}N$. We will however clarify this in the text, and include that additional studies are necessary to constrain this value.*

*We agree with the reviewer that this type of isotopic analysis would be interesting and think that future research regarding the substrates and reactions that produce oceanic RONO$_2$, as well as the processes that result in it influencing aerosol nitrate, would be useful to determine whether a $\delta^{15}N$ value of approximately -22‰ for RONO$_2$ as a source of aerosol nitrate is reasonable. This is however largely beyond the scope of our study. In light of this, we will not include text outlining the reactions involved in RONO$_2$ formation in section 4.2.2 to avoid repetition, as this RONO$_2$ formation mechanism is detailed in the introduction. We will add text to the conclusions suggesting that a mechanistic and isotopic understanding of these processes is needed from future studies as follows: "Additional research is needed to improve our mechanistic and isotopic understanding of surface ocean RONO$_2$ formation, flux, and conversion to aerosol nitrate in order to constrain the contribution of oceanic RONO$_2$ emissions to NO$_3^-$ formation in other ocean regions where this source has been invoked, such as the tropical Pacific (Kamezaki et al., 2019)."*

Line 467. I suggest replacing "unique" with distinct.

*As per the reviewer's suggestion, this sentence will now read: "Our observations across a large latitudinal gradient of the summertime Southern Ocean MBL suggest it is dominated by natural NOx sources with distinct isotopic signatures."*